# Modality-Decoupled Online Recursive Editing

**Siyuan Li** [1 2]   **Youyuan Zhang** [1 2]   **Fangming Liu** [2 3]   **Jing Li** ✉ [1]

## Abstract

Online model editing for multimodal large language models (MLLMs) requires assimilating a stream of corrections under tight compute and memory budgets. Yet editors developed for text-only LLMs often degrade on MLLMs: visually dominant activations skew the statistics that shape updates, causing *cross-modal conflict*, while sequential writes become entangled in a shared edit space and amplify long-horizon interference, causing *inter-edit interference*. To address these, we propose **M-ORE**, a modality-decoupled online recursive editor for lifelong MLLM adaptation. M-ORE is derived from a unified proximal-projection formulation and admits a closed-form update with a Sherman-Morrison recursion, yielding constant per-edit overhead. It maintains module-wise locality statistics for the text stack and the visual projector to avoid visually dominated update shaping and performs continual updates in a fixed orthogonal low-rank edit subspace via a Sherman-Morrison recursion to mitigate long-horizon interference. Experiments on multiple MLLM backbones and online editing benchmarks show that our M-ORE method consistently improves reliability, generality, and locality over strong baselines, while achieving favorable quality-efficiency scaling. Our code is publicly available at https://github.com/lab-klc/M-ORE.

## 1. Introduction

Large Language Models (LLMs) have become a foundation of natural language processing (Touvron et al., 2023; Zhao et al., 2023). Recent Multimodal Large Language Models (MLLMs) extend LLMs with visual perception and achieve

[1]Harbin Institute of Technology, Shenzhen, China. [2]Peng Cheng Laboratory, China. [3]Huazhong University of Science and Technology, China. Correspondence to: Jing Li <jingli.phd@hotmail.com>.

*Proceedings of the 43rd International Conference on Machine Learning*, Seoul, South Korea. PMLR 306, 2026. Copyright 2026 by the author(s).

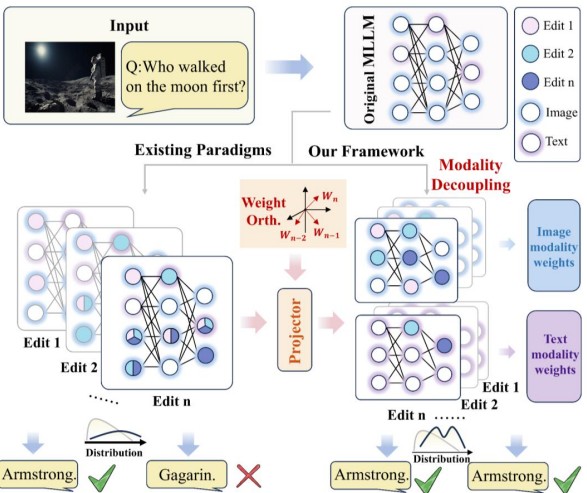

*Figure 1.* Overview of M-ORE and its contrast to existing online MLLM editing paradigms. M-ORE decouples vision/text updates and performs continual writes in a fixed orthogonal low-rank space.

strong performance on vision-language tasks (Lin et al., 2024; Zhu et al., 2026; Liang et al., 2026; Shi et al., 2026; Lu et al., 2026). However, their knowledge is encoded in parameters and thus remains static after training, reducing reliability as real-world facts evolve (Jiang et al., 2026). Since full retraining is costly, *model editing* provides an efficient alternative by updating knowledge with minimal parameter changes while preserving general capabilities (Meng et al., 2022; Fang et al., 2025).

Recent research in model editing has mainly focused on pure LLMs and achieved notable success (Mitchell et al., 2022a;b; Fang et al., 2025). However, directly applying text-oriented editors to MLLMs is often ineffective: the visual modality and its coupling with text introduce pronounced modality discrepancies (Shi et al., 2025; Chen et al., 2025b). Mainstream approaches (e.g., ROME and AlphaEdit (Meng et al., 2022; Fang et al., 2025)) typically optimize a global objective and implicitly assume that input representations share relatively homogeneous statistical properties; yet in MLLMs, visual representations often exhibit higher energy, thereby dominating the estimation of covariance and inducing *Cross-modal Conflict*. Moreover, practical systems require *online* editing under a stream of corrections (Cao et al., 2026), where parameter-modifying methods accumulate *inter-edit interference* and drift (Li & Chu, 2024), while

parameter-preserving methods incur edit-growing computation or storage (Chen et al., 2025a). As a result, existing MLLM editors rarely address both failure modes while retaining *constant-overhead* online updates.

To bridge this gap, we first conduct pilot analyses to diagnose the failure modes above. We observe that *cross-modal conflict* primarily arises from the mismatch in gradient-energy scales across modalities and *statistical heterogeneity*, whereas *inter-edit interference* stems from the competition between newly introduced edits and previously injected edits for representational capacity in the shared parameter space, causing *weight entanglement*. Figure 1 summarizes our high-level design and contrasts it with prior paradigms.

Driven by the preceding analysis, we propose Modality-decoupled Online Recursive Editing (M-ORE), a novel constant-overhead framework for lifelong adaptation in MLLMs. M-ORE maintains separate locality statistics for different edited modules (text layers vs. visual projector) to prevent *cross-modal conflict*, and performs continual writes in a *fixed orthogonal* low-rank subspace with a *Sherman-Morrison* grounded closed-form recursion. This design yields efficient online updates by adaptively suppressing updates on heavily-used coordinates in the fixed edit subspace, thereby reducing long-range *inter-edit entanglement*. Experiments on representative MLLM backbones and online multimodal editing benchmarks demonstrate that M-ORE achieves a better stability-plasticity trade-off than strong baselines under strict $O(1)$ per-edit overhead. Our main contributions are summarized as follows:

- We identify two key failure modes when applying LLM-based editors to MLLMs in the online setting: *cross-modal conflict* induced by modality scale mismatch and *inter-edit interference* induced by entangled sequential updates (Section 3).
- We propose M-ORE, a modality-decoupled editor with a Sherman–Morrison grounded closed-form recursion in a fixed orthogonal low-rank write space, enabling constant-overhead online editing (Section 4).
- We conduct extensive online editing experiments on multiple MLLM backbones and benchmarks, demonstrating that M-ORE consistently improves the stability-plasticity trade-off and achieves a quality-efficiency scaling compared with strong baselines (Section 5).

## 2. Related Work

### 2.1. Model Editing

**Model Editing for LLMs.** LLM editing methods are commonly grouped into *parameter-modifying* and *parameter-preserving* paradigms (Dai et al., 2022; Sinitsin et al., 2020; Zhang et al., 2024; Huang et al., 2023). Parameter-modifying editors directly update internal weights, includ-

ing (i) *locate-then-edit* methods such as ROME (Meng et al., 2022) and MEMIT (Meng et al., 2023), which localize knowledge-bearing components and apply closed-form low-rank updates; AlphaEdit (Fang et al., 2025) adds null-space constraints, and DeltaEdit (Cao et al., 2026) analyzes long-horizon error accumulation; and (ii) *meta-learning* editors such as KE (De Cao et al., 2021) and MEND (Mitchell et al., 2022a), which train hypernetworks to predict edit updates from gradients or error signals. Parameter-preserving approaches avoid changing the backbone, instead using retrieval/in-context prompting (e.g., IKE (Zheng et al., 2023)) or lightweight auxiliary models/modules (e.g., SERAC (Mitchell et al., 2022b), GRACE (Hartvigsen et al., 2023)) to locally override behavior. While effective for text-only settings, these techniques often degrade in multimodal regimes where visual inputs introduce substantial heterogeneity and noise (Chen et al., 2025b; Yu et al., 2024).

**Model Editing for MLLMs.** Editing MLLMs is still emerging (Cheng et al., 2023). Early studies mainly adapt LLM-based editors to multimodal benchmarks (He et al., 2024; Zhang et al., 2024); MMEdit systematically evaluates such adaptations and shows that naive transfer struggles with coupled vision-language representations (Cheng et al., 2023). Recent methods start to incorporate multimodal-specific designs: VisEdit (Chen et al., 2025b) uses attribution to identify and modify critical visual units; DualEdit (Shi et al., 2025) introduces gated dual-branch editing; LiveEdit (Chen et al., 2025a) targets lifelong editing via expert composition. Nevertheless, existing approaches typically incur edit-growing retrieval/composition costs or accumulate interference under sequential weight updates, and a principled framework that simultaneously mitigates *cross-modal conflict* and *inter-edit interference* under strict online constraints remains underexplored (Yao et al., 2023; Gu et al., 2024).

### 2.2. Online Editing

**Online Editing for LLMs.** Online (or sequential) editing extends single-step correction to a stream of updates, requiring models to address the plasticity-stability dilemma over time (Mitchell et al., 2022b; Jiang et al., 2025). In text-only LLMs, retrieval-based methods (Chen et al., 2024) mitigate catastrophic forgetting by storing edits in external memory, but incur inference latency that grows linearly with the number of edits(Wang et al., 2024b). Locate-then-edit approaches (Meng et al., 2022) are computationally efficient, yet often degrade severely in sequential settings.

**Online Editing for MLLMs.** In contrast, online editing for MLLMs remains under-explored (Chen et al., 2025a) and faces additional challenges induced by modality heterogene-

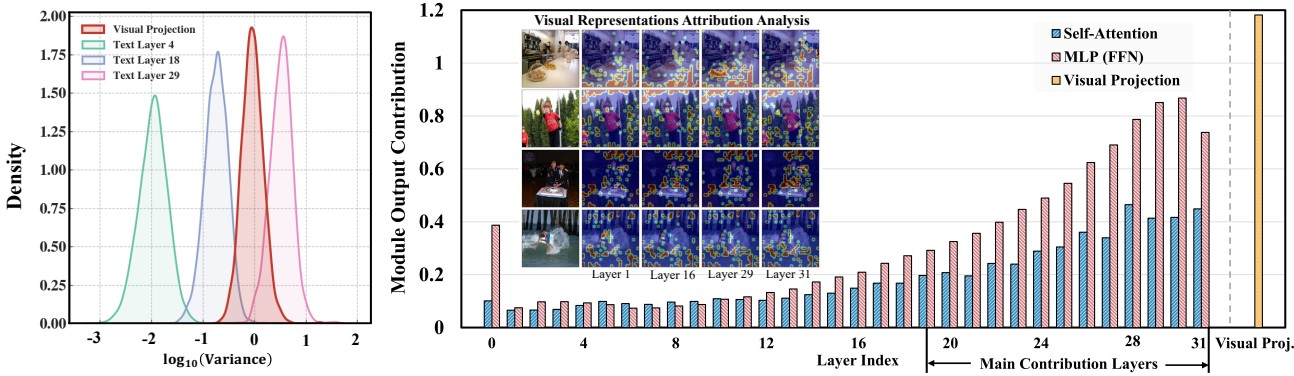

*Figure 2.* Cross-modal conflict caused by modality mismatch (BLIP2-OPT). **Left:** log-variance density indicates higher-energy visual features than text activations. **Right:** attribution analysis shows visually dominated contributions, implying that shared global statistics bias updates toward the visual subspace and weaken textual preservation. Results for other MLLMs are provided in the Appendix B.1.

ity (Cheng et al., 2023; Chen et al., 2025b; Shi et al., 2025). Current solutions largely rely on *parameter-preserving*. For example, LiveEdit (Chen et al., 2025a) proposes a mixture-of-experts framework that generates an edit-specific LoRA expert for each sample and routes queries at inference time. However, such approaches incur substantial memory overhead as the number of edits increases (Shi et al., 2025; Thede et al., 2025). Crucially, *parameter-modifying* online editing for MLLMs remains largely unexplored (Li et al., 2025; Zhang et al., 2025; Deng et al., 2025).

## 3. Analysis on Online Sequential Editing

### 3.1. Preliminary

**Multimodal Language Model.** We consider an MLLM $f_\theta$ with a vision encoder $\mathcal{E}_v$, projector $\mathcal{P}$, and LLM $\mathcal{M}$. Given $(x^v, x^q)$, where $x_t^v$ is the visual input, $x_t^q$ is the text query, and the model autoregressively predicts $p(y \mid x^v, x^q)$ from embeddings $H = [\mathcal{P}(\mathcal{E}_v(x^v)), \mathrm{Embed}(x^q)]$. For transformer layer $l$, we adopt the standard residual decomposition (Meng et al., 2022; Fang et al., 2025):

$$h^l = h^{l-1} + a^l + m^l, \quad m^l = W_{\mathrm{out}}^l \sigma\big(W_{\mathrm{in}}^l \gamma(h^{l-1} + a^l)\big), \quad (1)$$

where $a^l$ and $m^l$ denote the attention and FFN outputs, $\gamma(\cdot)$ is LayerNorm, and $\sigma(\cdot)$ is a nonlinearity. Following the FFN key-value view (Geva et al., 2021), we define

$$k^l \triangleq \sigma\big(W_{\mathrm{in}}^l \gamma(h^{l-1} + a^l)\big), \quad v^l \triangleq m^l = W_{\mathrm{out}}^l k^l. \quad (2)$$

Many parameter-modifying editors rewrite knowledge by updating $W_{\mathrm{out}}^l$; we denote it as $W$ when clear.

**Online Editing in MLLMs.** We study online editing of an MLLM $f_\theta$ that maps a multimodal input $x = (x^v, x^q) \in \mathcal{X}$ to an autoregressive output distribution over text. Following the FFN key-value view, we consider *parameter-modifying* edits that update a selected FFN output matrix $W$ while keeping the remaining parameters fixed, i.e., $\theta = (\theta_0, W)$

with $\theta_0$ frozen. At step $t$, an edit request is a target pair $e_t = (x_t^e, y_t^e)$ such that $y_t^e \neq f_{(\theta_0, W_{t-1})}(x_t^e)$. An editor ME produces an additive update with bounded per-edit cost:

$$\Delta W_t = \mathrm{ME}\big(f_{(\theta_0, W_{t-1})}, x_t^e, y_t^e\big), \quad W_t = W_{t-1} + \Delta W_t, \quad (3)$$

so $f_{\theta_t} = f_{(\theta_0, W_t)}$ and $W_t = W_0 + \sum_{i=1}^t \Delta W_i$.

A successful online editor should satisfy three criteria (Cheng et al., 2023; Chen et al., 2025a): *Reliability* (correct on the edited request), *Generality* (holds under semantically equivalent text/visual variants), and *Locality* (minimal side effects on unrelated inputs). We quantify locality by the distributional shift between $f_{\theta_t}$ and $f_{\theta_{t-1}}$ on an irrelevant set $\mathcal{U}_t$:

$$\mathbb{E}_{x \sim \mathcal{U}_t}\Big[ \exp\big( - \mathrm{KL}\big(p_{\theta_t}(\cdot|x) \,\|\, p_{\theta_{t-1}}(\cdot|x)\big)\big)\Big], \quad (4)$$

where $p_\theta(\cdot|x)$ is the next-token distribution under the same decoding prefix. Full metric definitions are in Appendix A.2. We next present pilot analyses to motivate our design.

### 3.2. Cross-modal Conflict

We identify *cross-modal conflict* as a key failure mode when applying text-oriented editors to MLLMs: visual representations typically have much larger scale and can dominate the second-order statistics used for preconditioning or constraints, thereby biasing edits toward the visual subspace [1].

**Statistical heterogeneity.** We compute diagonal variances of textual hidden states across transformer layers and compare them with the projected visual representations. Figure 2 reveals a clear scale gap: variance magnitudes vary across text layers, while the visual projection consistently stays at a higher scale than early text layers. As a result, pool-

---

[1] Estimating full cross-modal covariance is $O(d^2)$ and impractical at MLLM scale; we therefore use diagonal variance and layer-wise energy as simple second-order proxies, which already reveal the scale mismatch that biases pooled statistics.

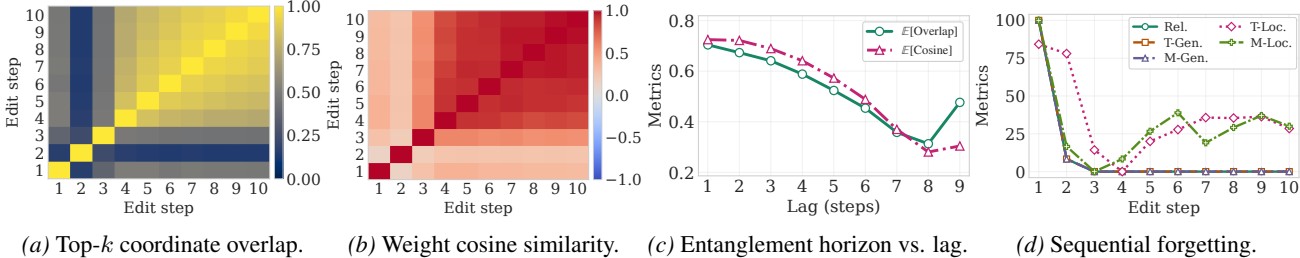

*(a)* Top-$k$ coordinate overlap.    *(b)* Weight cosine similarity.    *(c)* Entanglement horizon vs. lag.    *(d)* Sequential forgetting.

*Figure 3.* Inter-edit interference induced by MEND on BLIP2-OPT over 10 online sequential edits on the E-IC test set.

ing statistics across modalities is prone to visual-dominated estimates and cross-modal contamination.

**Energy dominance and propagation.** We further measure layer-wise output energy averaged over the same edits. Figure 2 shows that the visual projection contributes the largest energy, and the MLP branch increasingly dominates in higher layers; deeper blocks also attend more strongly to salient image regions. These patterns suggest that strong visual signals are amplified and propagated through shared blocks. Taken together, these results motivate maintaining *separate* second-order statistics for different edited modules (text layers vs. visual projection), rather than sharing a single statistic across modalities.

### 3.3. Inter-edit Interference

Beyond single-step reliability, *online* MLLM editing must absorb a stream of updates without overwriting previously injected knowledge. We observe a common failure mode, *inter-edit interference*, where successive edits become increasingly entangled in a shared parameter subspace, leading to accumulated drift and forgetting.

**Edits collapse into an "edit core".** At each step $t$, we extract the effective update $\Delta W_t$ and compute pairwise overlap of the top-$k$ updated coordinates. Figure 3a shows a clear transition: early edits activate diverse coordinates, whereas later edits repeatedly modify a compact, stable subset, indicating collapse into a shared *edit core*.

**Entanglement is predominantly co-directional.** We further measure directional coupling via cosine similarity. Figure 3b reveals mostly positive correlations that become strongly aligned in the late-stage block, suggesting that interference is driven by co-directional accumulation within the reused edit core rather than push–pull cancelations. Consistently, Figure 4 shows that MLLM hidden-state distributions drift after only a few edits.

**Long-range coupling explains forgetting.** Aggregating the above pairwise statistics by lag $\tau$ yields an *entanglement horizon* (Figure 3c): both overlap and cosine decay slowly and remain non-trivial for distant edits. This long-range coupling provides a parameter-level explanation for sequential degradation: as new edits continue to reuse and move along

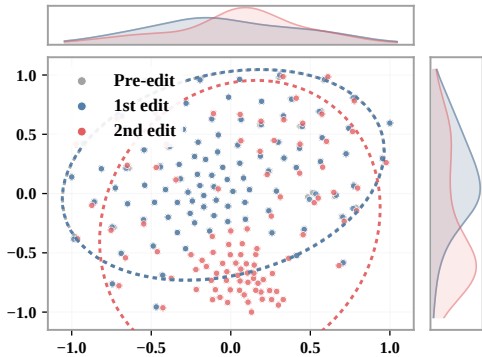

*Figure 4.* Sequential visualization of MEND-induced hidden-state shifts on BLIP2-OPT across 100 consecutive locality samples.

the same edit-core subspace, earlier edits are progressively overwritten, matching the forgetting trends in Figure 3d. Motivated by these observations, sequential updates should be geometrically de-entangled. We thus restrict edits to a fixed orthogonal low-rank write space and apply a recursive rule with an implicit orthogonalization bias, discouraging repeated reuse of heavily-occupied coordinates and mitigating collapse and drift.

## 4. Methodology

### 4.1. Unified Proximal Projection Principle

Existing MLLM editors are mostly adapted from *locate-then-edit* or *meta-learning* paradigms, both of which are fragile in the online multimodal regime: reliable localization is costly, while meta-training is expensive and sensitive to deployment-time shifts. We instead adopt a *unified optimization* view: each online edit should (i) fit the current request (reliability/generality) while (ii) minimizing side effects on unrelated inputs (locality). At step $t$, let $W_{t-1}$ denote the editable parameters and $g_t = \nabla_{W_{t-1}} \mathcal{L}_{\text{edit}}$ be the edit gradient, where $\mathcal{L}_{\text{edit}}$ denotes the standard editing loss (See Appendix A.3). We compute the update via a single *proximal projection*:

$$\Delta W_t = \arg\min_{\Delta W} \|\Delta W + \eta g_t\|_F^2 + \text{Tr}\left(\Delta W\, C_{t-1}\, \Delta W^\top\right), \quad (5)$$

where the first term is a one-step descent surrogate for fitting the edit, and the quadratic form $C_{t-1}$ penalizes updates

along directions that are important for locality. The locality statistic $C_{t-1}$ is accumulated online from step 1 through step $t-1$, using a fixed-size *step-only* context set $\mathcal{B}_s$ at each step $s$ (e.g., edit batch and a small locality sample at step $s$):

$$C_{t-1} = C_0 + \sum_{s=1}^{t-1} \sum_{x \in \mathcal{B}_s} k_s(x)k_s(x)^\top, \quad C_0 = \mathbf{0}. \quad (6)$$

where $k_s(x)$ is the FFN key induced by multimodal input $x$ at the edited module. This projection view is closely related to the quadratic target-matching formulation used by *locate-then-edit* methods (Appendix C).

## 4.2. Drift-Free Low-Rank Coordinate Subspace

Maintaining $C_{t-1}$ in the full space is infeasible online. We restrict edits to a low-rank write interface with a *time-invariant* coordinate system. For each edited layer $l \in \mathcal{L}$:

$$\widetilde{W}^{(l)} = W^{(l)} + \Delta W^{(l)}, \quad \Delta W^{(l)} = B^{(l)}A^{(l)}, \quad (7)$$

where $A^{(l)} \in \mathbb{R}^{r \times d}$ defines a fixed write coordinate system and $B^{(l)} \in \mathbb{R}^{d_{\text{out}} \times r}$ is updated online.

**Frozen orthogonal coordinates.** We orthogonally initialize and *freeze* $A^{(l)}$ such that its rows are orthonormal:

$$A^{(l)}A^{(l)\top} = I_r, \quad A^{(l)} \text{ frozen}, \quad B^{(l)} \text{ updated (online).} \quad (8)$$

This fixed coordinate system enables us to carry the locality quadratic constraint $C_{t-1}$ in Eq. (5) into a stable low-dimensional statistic, which can be accumulated online.

**Steady-space locality features.** Given buffer keys $k_{t-1}^{(l)}(x) \in \mathbb{R}^d$, we project them into the steady coordinates,

$$z_{t-1}^{(l)}(x) = A^{(l)}k_{t-1}^{(l)}(x) \in \mathbb{R}^r. \quad (9)$$

The full projected second-order statistic would be $\sum_{x \in \mathcal{B}_{t-1}^{(l)}} z_{t-1}^{(l)}(x)z_{t-1}^{(l)}(x)^\top$. To keep constant overhead, we summarize the step context with a rank-one sketch obtained by masked pooling over the relevant token positions: $\bar{z}_{t-1}^{(l)} \triangleq \text{Mean}\big(\{z_{t-1}^{(l)}(x) \mid x \in \mathcal{B}_{t-1}^{(l)}\}\big) \in \mathbb{R}^r$. We then maintain a compact steady-space locality matrix:

$$S_{t-1}^{(l)} = S_0^{(l)} + \sum_{s=1}^{t-1} \bar{z}_s^{(l)}(\bar{z}_s^{(l)})^\top, \quad S_0^{(l)} = \lambda I_r, \ \lambda > 0. \quad (10)$$

where $\lambda$ provides numerical stability and sets the plasticity–locality trade-off. A formal derivation of the induced steady-space quadratic form is provided in Appendix D.1.

## 4.3. Steady-Subspace Recursive Least Squares Update

We now instantiate Eq. (5) in the steady space with $B^{(l)}$ as the editable variables. Let $G_t^{(l)} = \nabla_{B_{t-1}^{(l)}} \mathcal{L}_{\text{edit}}$ be the corresponding gradient at edit step $t$. The current per-layer update is computed using only the statistics accumulated before step $t$:

$$\arg \min_{\Delta B^{(l)}} \ \|\Delta B^{(l)} + \eta G_t^{(l)}\|_F^2 + \text{Tr}\Big(\Delta B^{(l)} S_{t-1}^{(l)} (\Delta B^{(l)})^\top\Big), \quad (11)$$

whose unique minimizer is the closed-form write:

$$\Delta B_t^{(l)} = -\eta \, G_t^{(l)}\big(I + S_{t-1}^{(l)}\big)^{-1} \triangleq -\eta \, G_t^{(l)} P_{t-1}^{(l)}. \quad (12)$$

Here, $P_{t-1}^{(l)} \triangleq (I + S_{t-1}^{(l)})^{-1}$ serves as the steady-space preconditioner for the current edit at step $t$. Since $S_{t-1}^{(l)}$ accumulates outer products of previous steady features, $P_{t-1}^{(l)}$ suppresses updates on coordinates that have been repeatedly activated by past edits, yielding an implicit de-entangling bias across edits. After applying the current edit, the current step feature $\bar{z}_t^{(l)}$ is efficiently inserted into the recursive statistic via the Sherman–Morrison lemma, avoiding direct matrix inversion and yielding the preconditioner for the next edit step $t+1$:

$$P_t^{(l)} = P_{t-1}^{(l)} - \frac{P_{t-1}^{(l)} \bar{z}_t^{(l)} (\bar{z}_t^{(l)})^\top P_{t-1}^{(l)}}{1 + (\bar{z}_t^{(l)})^\top P_{t-1}^{(l)} \bar{z}_t^{(l)}}, \quad P_0^{(l)} = \frac{1}{1+\lambda} I_r. \quad (13)$$

Thus, $P_{t-1}^{(l)}$ is used to compute $\Delta B_t^{(l)}$, while the updated $P_t^{(l)}$ is cached and used only from step $t+1$ onward. Each active layer stores only $P_t^{(l)} \in \mathbb{R}^{r \times r}$ and updates it in $O(r^2)$ time, yielding constant per-edit overhead. Derivations of Eq. (12) and Eq. (13) are in Appendix D.

## 4.4. Implications of the Steady-Space Recursion

Steady-space RLS rule in Eq. (12)–Eq. (13) already implies the key design principles needed for online MLLM editing.

**How Does M-ORE Resolve *Cross-Modal Conflict*?** Cross-modal conflict arises when heterogeneous modules (visual projector vs. text layers) are forced to share statistics (Section 3.2). Our formulation avoids this issue by maintaining a separate preconditioner $P^{(l)}$ for each edited module, so that vision-side and text-side statistics are fully decoupled. High-energy visual features are therefore confined to the projector statistics, preventing contamination of text-side updates. When forming $\bar{z}^{(l)}$, we further apply modality-specific masks and pool only the relevant token positions, preventing mixed-token summaries inside MLLMs.

**How Does M-ORE Mitigate *Inter-Edit Interference*?** Inter-edit interference is driven by repeatedly reusing a shared *edit core* (Section 3.3). In our fixed orthogonal write coordinates (Eq. (8)), the accumulation $S_{t-1}^{(l)} = \lambda I_r + \sum_{i \leq t-1} \bar{z}_i^{(l)} (\bar{z}_i^{(l)})^\top$ increases the penalty on frequently used coordinates, and the resulting preconditioner $P_{t-1}^{(l)}$ adaptively suppresses writes into those occupied coordinates.

*Table 1.* Online editing results on E-VQA and E-IC for BLIP2-OPT and LLaVA-v1.5 under different edit horizons. "Rel.", "T/M-Gen.", and "T/M-Loc." abbreviate *Reliability*, *Generality*, and *Locality* (for text/modal evaluations), respectively. The subscript of each method (e.g., $_1$, $_{100}$) denotes the number of online edits performed. Rows shaded in light purple indicate *parameter-modifying* methods. Due to space limitations, complete results are provided in Appendix B.2.

| Model | Methods | E-VQA | | | | | | E-IC | | | | | |
|---|---|---|---|---|---|---|---|---|---|---|---|---|---|
| | | Rel.$^\uparrow$ | T-Gen.$^\uparrow$ | M-Gen.$^\uparrow$ | T-Loc.$^\uparrow$ | M-Loc.$^\uparrow$ | Avg.$^\uparrow$ | Rel.$^\uparrow$ | T-Gen.$^\uparrow$ | M-Gen.$^\uparrow$ | T-Loc.$^\uparrow$ | M-Loc.$^\uparrow$ | Avg.$^\uparrow$ |
| BLIP2-OPT | FT-L$_1$ | 100.00 | 100.00 | 60.00 | 94.74 | 100.00 | 90.95 | 96.77 | 95.02 | 90.72 | 90.05 | 68.27 | 88.16 |
| | FT-M$_1$ | 100.00 | 96.67 | 63.33 | 100.00 | 73.33 | 86.67 | 100.00 | 100.00 | 76.92 | 100.00 | 24.79 | 80.34 |
| | MEND$_1$ | 100.00 | 100.00 | 100.00 | 60.61 | 33.33 | 78.79 | 100.00 | 100.00 | **100.00** | 84.21 | 100.00 | 96.84 |
| | AlphaEdit$_1$ | 60.00 | 50.00 | 40.00 | 91.64 | 73.33 | 62.99 | 30.77 | 30.77 | 30.77 | 89.47 | 100.00 | 56.35 |
| | SERAC$_1$ | 100.00 | 100.00 | 100.00 | 89.47 | 80.00 | 93.89 | 97.38 | 95.52 | 86.11 | 100.00 | 63.82 | 88.57 |
| | IKE$_1$ | 100.00 | 100.00 | 100.00 | 52.63 | 13.33 | 73.19 | 100.00 | 100.00 | 100.00 | 63.16 | 10.00 | 74.63 |
| | LiveEdit$_1$ | 92.68 | 92.33 | 89.25 | **100.00** | 95.57 | 93.97 | 80.67 | 80.67 | 77.79 | 100.00 | 98.09 | 87.44 |
| | **M-ORE$_1$** | **100.00** | **100.00** | **100.00** | 94.74 | **100.00** | **98.95** | **100.00** | **100.00** | 93.75 | **100.00** | **100.00** | **98.75** |
| | FT-L$_{100}$ | 26.00 | 18.00 | 11.00 | 86.28 | 53.12 | 38.88 | 79.45 | 71.69 | 57.82 | 92.23 | 55.18 | 71.27 |
| | FT-M$_{100}$ | 58.40 | 50.85 | 43.69 | **100.00** | 46.85 | 59.96 | 67.18 | 62.17 | 51.63 | **100.00** | 9.81 | 58.16 |
| | MEND$_{100}$ | 1.00 | 1.00 | 1.00 | 90.42 | 77.20 | 34.12 | 0.00 | 0.00 | 0.00 | 46.96 | 54.05 | 20.20 |
| | AlphaEdit$_{100}$ | 36.83 | 28.50 | 26.78 | 86.35 | 69.97 | 49.69 | 27.94 | 28.57 | 26.69 | 90.52 | 78.83 | 50.51 |
| | SERAC$_{100}$ | 88.03 | 85.18 | 88.03 | 89.95 | 37.60 | 77.76 | 74.02 | 63.79 | 56.93 | 77.52 | 42.94 | 63.04 |
| | IKE$_{100}$ | 83.53 | 83.00 | 84.72 | 74.63 | 6.37 | 66.45 | 81.18 | 71.13 | **84.30** | 79.90 | 11.93 | 65.69 |
| | LiveEdit$_{100}$ | 91.83 | 91.16 | 85.02 | 99.31 | **92.78** | 92.02 | 74.63 | 74.33 | 61.95 | 96.77 | 95.34 | 80.60 |
| | **M-ORE$_{100}$** | **96.05** | **92.05** | **94.06** | 91.77 | 88.85 | **92.56** | **83.14** | **78.17** | 60.08 | 95.51 | **95.58** | **82.49** |
| LLaVA-v1.5 | FT-L$_1$ | 95.66 | 99.00 | 81.84 | 89.38 | 89.98 | 91.17 | 100.00 | 100.00 | 89.80 | 91.67 | 28.01 | 81.89 |
| | FT-M$_1$ | 95.00 | 95.00 | 79.67 | 100.00 | 85.83 | 91.10 | 100.00 | 100.00 | 76.47 | 100.00 | 26.11 | 80.52 |
| | MEND$_1$ | 95.53 | 95.53 | 83.79 | 74.82 | 59.65 | 81.86 | 96.81 | 97.65 | **96.44** | 94.74 | 100.00 | 97.13 |
| | AlphaEdit$_1$ | 72.57 | 72.57 | 70.67 | 88.04 | 96.67 | 80.10 | 47.06 | 47.06 | 52.94 | 100.00 | 100.00 | 69.41 |
| | SERAC$_1$ | 90.00 | 90.00 | 60.00 | 100.00 | 23.33 | 72.67 | 88.61 | 90.17 | 81.45 | 98.24 | 50.83 | 81.86 |
| | IKE$_1$ | 50.00 | 50.00 | 50.00 | 58.33 | 15.00 | 44.67 | 94.12 | 94.12 | 94.12 | 62.50 | 12.50 | 71.47 |
| | LiveEdit$_1$ | 93.36 | 93.67 | **87.91** | 100.00 | 100.00 | 94.99 | 82.33 | 82.33 | 80.67 | 100.00 | 100.00 | 89.07 |
| | **M-ORE$_1$** | **100.00** | **100.00** | 85.12 | **100.00** | **100.00** | **97.02** | **100.00** | **100.00** | 94.12 | **100.00** | **100.00** | **98.82** |
| | FT-L$_{100}$ | 74.55 | 66.59 | 66.14 | 84.15 | 65.71 | 71.43 | 86.00 | 82.11 | 78.15 | 77.13 | 11.33 | 66.94 |
| | FT-M$_{100}$ | 85.67 | 81.75 | 67.03 | **100.00** | 46.19 | 76.13 | 84.57 | 80.05 | 62.88 | **100.00** | 8.32 | 67.16 |
| | MEND$_{100}$ | 1.09 | 1.09 | 1.01 | 75.33 | 61.67 | 28.04 | 0.11 | 0.08 | 0.04 | 25.52 | 31.33 | 11.42 |
| | AlphaEdit$_{100}$ | 71.33 | 70.93 | 70.67 | 88.45 | 77.67 | 75.81 | 51.98 | 48.21 | 47.23 | 92.59 | 56.90 | 59.38 |
| | SERAC$_{100}$ | 87.71 | 85.03 | 67.13 | 92.27 | 20.83 | 70.59 | 73.38 | 71.00 | 59.72 | 72.88 | 25.85 | 60.57 |
| | IKE$_{100}$ | 38.87 | 35.85 | 39.40 | 46.22 | 11.24 | 34.32 | 78.78 | 77.37 | 78.07 | 53.86 | 12.88 | 60.19 |
| | LiveEdit$_{100}$ | 90.22 | 91.39 | 81.49 | 98.05 | **95.27** | 91.28 | 78.49 | 78.77 | 65.50 | 98.77 | **96.76** | 83.66 |
| | **M-ORE$_{100}$** | **97.43** | **94.30** | **85.40** | 96.32 | 91.90 | **93.07** | **94.64** | **93.27** | 78.66 | 97.39 | 89.71 | **90.73** |

Consequently, $\Delta B_t^{(l)} = -\eta G_t^{(l)} P_{t-1}^{(l)}$ discourages persistent reuse of the same edit subspace, reducing long-horizon entanglement with constant overhead.

# 5. Experiments

We conduct experiments to address the following questions:

- **RQ1:** How does M-ORE compare with baselines in online MLLM editing, particularly in alleviating *cross-modal conflict* and *inter-edit interference*?
- **RQ2:** Does M-ORE preserve the edited model's general capabilities on standard generalization evaluations?
- **RQ3:** What are the time and space complexities of M-ORE for each online edit compared to the baselines? Concretely, how does M-ORE achieve a better quality–efficiency trade-off under the same settings?

- **RQ4:** Can M-ORE effectively prevent shifts in the distribution of hidden representations after editing?

## 5.1. Experiment Setup

We briefly introduce the datasets, metrics, backbones and baselines, with full experimental setup and orthogonal-basis implementation details in Appendices A and A.6.

**Datasets & Metrics.** Following (Cheng et al., 2023), we evaluate online editing on E-VQA (Editing VQA) and E-IC (Editing Image Caption), where E-IC requires finer-grained visual grounding. We report *Reliability*, *Generality*, and *Locality* (Section 3.1), and further decompose Generality/Locality into text- and image-conditioned evaluations.

**MLLM Backbones & Baseline Editors.** We use two representative MLLM backbones with distinct architec-

*Table 2.* Ablation study of M-ORE design choices on E-IC task for LLaVA-v1.5. $\Delta$ indicates the change in Average score.

| Model | Variant | E-IC | | | | | | $\Delta$ |
|---|---|---|---|---|---|---|---|---|
| | | Rel.$^\uparrow$ | T-Gen.$^\uparrow$ | M-Gen.$^\uparrow$ | T-Loc.$^\uparrow$ | M-Loc.$^\uparrow$ | Avg.$^\uparrow$ | |
| LLaVA-v1.5 | **M-ORE$_1$** | 100.00 | 100.00 | 94.12 | 100.00 | 100.00 | 98.82 | – |
| | w/o freezing $A^{(l)}{}_1$ | 97.98 | 97.21 | 85.24 | 100.00 | 81.36 | 92.35 | $-6.47$ |
| | w/o pooling$_1$ | 100.00 | 100.00 | 96.67 | 100.00 | 100.00 | 99.33 | $+0.51$ |
| | **M-ORE$_{100}$** | 94.64 | 93.27 | 78.66 | 97.39 | 89.71 | 90.73 | – |
| | w/o freezing $A^{(l)}{}_{100}$ | 83.70 | 82.92 | 71.25 | 80.86 | 44.43 | 72.63 | $-18.10$ |
| | w/o pooling$_{100}$ | 95.79 | 94.84 | 80.20 | 98.27 | 93.05 | 92.43 | $+1.70$ |

*Table 3.* Effect of different write-space basis initializations on E-IC with LLaVA-v1.5. We compare Gaussian, Xavier, data-driven, and the proposed orthogonal initialization under short- and long-horizon online editing. The data-driven basis is initialized using activation statistics collected from the training set

| Basis Choice | Rel. | T-Gen. | M-Gen. | T-Loc. | M-Loc. |
|---|---|---|---|---|---|
| Gaussian$_1$ | 76.47 | 76.47 | 64.71 | 100.00 | 100.00 |
| Xavier$_1$ | 5.88 | 5.88 | 5.88 | 100.00 | 100.00 |
| Data-driven$_1$ | 100.00 | 100.00 | 94.12 | 91.67 | 100.00 |
| **Orthogonal$_1$** | **100.00** | **100.00** | **94.12** | **100.00** | **100.00** |
| Gaussian$_{100}$ | 71.43 | 70.76 | 67.04 | 95.71 | 70.76 |
| Xavier$_{100}$ | 8.94 | 8.85 | 9.13 | 96.02 | 85.50 |
| Data-driven$_{100}$ | 90.07 | 85.79 | 72.29 | 86.59 | 83.02 |
| **Orthogonal$_{100}$** | **94.64** | **93.27** | **78.66** | **97.39** | **89.71** |

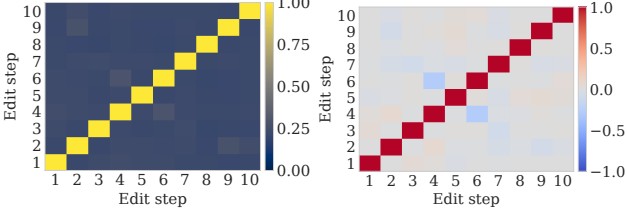

*Figure 5.* Inter-edit interference statistics of M-ORE on BLIP2-OPT over 10 online sequential edits.

tures and scales: BLIP2-OPT (2.7B) (Li et al., 2023) and LLaVA-v1.5 (7B) (Liu et al., 2024). Since dedicated online MLLM editors remain limited, we follow MMEdit-style evaluation (Cheng et al., 2023) and adapt widely-used LLM editors for comparison. We group baselines into *parameter-modifying* methods (FT-L/FT-M (Cheng et al., 2023), MEND (Mitchell et al., 2022a), AlphaEdit (Fang et al., 2025)) and *parameter-preserving* methods (IKE (Zheng et al., 2023), SERAC (Mitchell et al., 2022b), LiveEdit (Chen et al., 2025a)).

### 5.2. Comprehensive Performance Comparison (RQ1)

**Main Results.** Table 1 summarizes online editing results on multiple MLLM backbones, where long edit streams expose long-horizon robustness beyond single-edit success. Overall, existing baselines exhibit two recurring failure patterns. *Parameter-modifying* methods, such as FT variants and MEND, can achieve strong single-edit reliability but

tend to accumulate drift under sequential edits, leading to severe degradation at $t=100$. For example, MEND collapses to nearly zero reliability and generality on both backbones after long-horizon editing. AlphaEdit, while effective for text-only LLMs, remains brittle in MLLMs, likely due to unreliable multimodal localization and second-order statistics under modality heterogeneity. *Parameter-preserving* methods, such as SERAC, IKE, and LiveEdit, better avoid direct weight drift, but still struggle to maintain stable multimodal behavior as the edit horizon grows.

In contrast, M-ORE consistently maintains a stronger stability-plasticity trade-off under long edit horizons. Under $t=100$ edits, M-ORE$_{100}$ achieves the best average performance across both backbones and both tasks. On BLIP2-OPT, M-ORE$_{100}$ improves over LiveEdit$_{100}$ by $+0.54$ average points on E-VQA and $+1.89$ average points on E-IC. In particular, it improves E-IC reliability from 74.63 to 83.14 and text generality from 74.33 to 78.17, while keeping locality comparable. On LLaVA-v1.5, the gains are more pronounced: M-ORE$_{100}$ improves the average score over LiveEdit$_{100}$ by $+1.79$ points on E-VQA and $+7.07$ points on E-IC. For E-IC, it substantially improves reliability from 78.49 to 94.64, text generality from 78.77 to 93.27, and multimodal generality from 65.50 to 78.66, demonstrating stronger long-horizon edit retention and generalization. Moreover, Figure 5 shows no late-stage "edit-core" collapse for M-ORE: cross-step top-$k$ overlaps remain low and cosine couplings stay near zero, indicating more disentangled sequential updates and reduced long-range drift/forgetting. **For additional experimental results, including sensitivity analyses, extension to vision-side editing, and case studies, please refer to Appendix B.4, B.5, and B.6.**

**Ablation Studies.** We conduct ablation experiments to validate M-ORE's key design choices, including the drift-free orthogonal write space and the constant-cost sketch/pooling scheme for estimating locality statistics. Specifically, we consider the following variants:

- **w/o freezing** $A^{(l)}$: This variant updates the write basis $A^{(l)}$ online jointly with $B^{(l)}$, rather than keeping $A^{(l)}$ fixed as in Eq. (8). It removes the drift-free coordinate

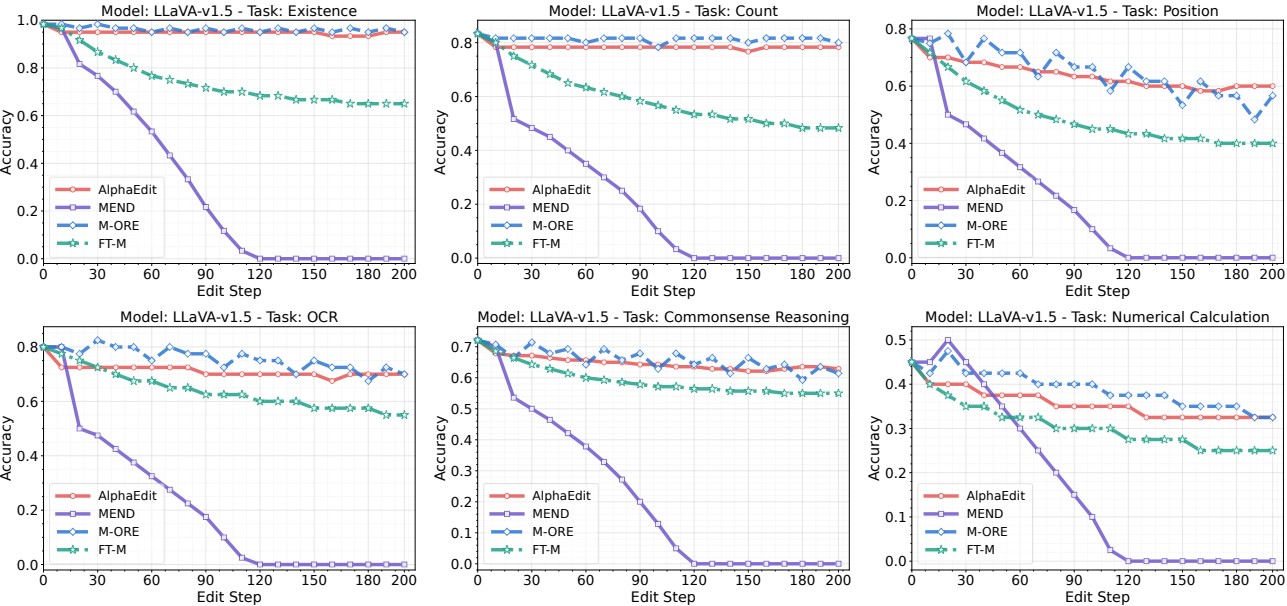

*Figure 6.* accuracy of the post-edited LLaVA-v1.5 (7B) on six tasks used for MLLM general capability testing.

constraint and allows the low-rank write space to change over the edit stream.

- **w/o pooling**: This variant replaces the rank-one sketch in Eq. (10) with the full projected statistic $\sum_{x \in \mathcal{B}_{t-1}^{(l)}} z_{t-1}^{(l)}(x) z_{t-1}^{(l)}(x)^{\top}$, and then updates $P_t^{(l)}$.

Table 2 shows that *freezing* the orthogonal basis $A^{(l)}$ is critical for long-horizon stability: removing it consistently reduces the average performance, and the degradation grows with the edit horizon, most notably at $t=100$. This indicates that a stable coordinate system is necessary to prevent sequential drift and mitigate accumulated inter-edit interference. In contrast, removing masked pooling when forming $\bar{z}_t^{(l)}$ does not degrade performance and can slightly improve the average score, suggesting that our rank-one sketch is a robust constant-overhead approximation to the full projected statistic rather than a fragile heuristic.

**Basis Initialization Analysis.** To examine the robustness of the steady-subspace design more explicitly, we compare different initializations of the fixed basis $A^{(l)}$. As shown in Table 3, random initializations such as Gaussian and Xavier are substantially less stable, especially under long edit horizons. Although the data-driven basis improves over random initialization, it still falls behind the orthogonal basis. In contrast, the proposed orthogonal initialization achieves the best overall performance at both $t=1$ and $t=100$, demonstrating that an orthonormal and drift-free coordinate system provides a more reliable low-rank write space for editing. We further test five random seeds for the orthogonal basis. As shown in Table 4, M-ORE remains highly stable across seeds, with average absolute deviations of only 0.07 on E-VQA and 0.13 on E-IC relative to the default seed 42.

*Table 4.* Seed sensitivity of the orthogonal basis initialization on LLaVA-v1.5 with $t=100$ online edits.

| Metric | 40 | 41 | **42** | 43 | 44 |
|---|---|---|---|---|---|
| E-VQA Avg. | 93.16 | 93.11 | 93.07 | 93.11 | 93.16 |
| E-IC Avg. | 90.40 | 90.87 | 90.73 | 90.87 | 90.69 |

### 5.3. General Capability Evaluation (RQ2)

To assess whether editing preserves general-purpose multimodal abilities, we evaluate the post-edited LLaVA-v1.5 on representative categories from the MME benchmark (Fu et al., 2023). Since MME contains 14 categories, we report six representative ones here and defer the remaining results to Appendix B.3. The selected categories include:

- **Existence**: tests whether a queried object/concept is present in the image.
- **Count**: evaluates counting by verifying the queried number of target objects.
- **Position**: measures spatial understanding of object locations and relative relations.
- **OCR**: assesses text recognition and grounding for image-based questions.
- **Commonsense Reasoning**: probes image-grounded commonsense inference beyond literal perception.
- **Numerical Calculation**: evaluates image-grounded arithmetic based on numbers/formulas in the image.

Figure 6 shows that M-ORE largely preserves general capabilities after editing, with stability comparable to the unedited model and AlphaEdit. The advantage is most evident on reasoning-heavy tasks (e.g., OCR and Numerical Calculation), where M-ORE avoids the progressive degra-

*Table 5.* Per-edit computational and memory complexities and their dependence on edit stream length $t$. Time/edit accounts for editor-specific updates and statistic maintenance, excluding the shared forward/backward cost for computing edit gradients. Here $r \ll d$, $|\mathcal{L}|$ denotes the number of edited layers, and $p_{\text{mem}}$ denotes the size of an edit-specific stored item. We do not include meta-learning editors since they require an additional training stage.

| Method | Time / edit | Memory | $t$-dep. |
|---|---|---|---|
| **M-ORE (ours)** | $O\big(|\mathcal{L}|\, d_{\text{out}} r^2\big)$ | $O\big(|\mathcal{L}|\, d_{\text{out}} r\big)$ | $O(1)$ |
| Locate-then-edit | $O(d^3)$ or $O(d^2)$ | $O(d^2)$ | $O(1)$ |
| Null-space constraint | $O(d\, t^2)$ | $O(d\, t)$ | ↑ |
| Parameter-preserving | $O(t)$ or $O(\log t)$ | $O(t\, p_{\text{mem}})$ | ↑ |
| Naive finetuning | $O\big(|\mathcal{L}|\, d_{\text{out}} r\big)$ | $O\big(|\mathcal{L}|\, d_{\text{out}} r\big)$ | $O(1)$ |

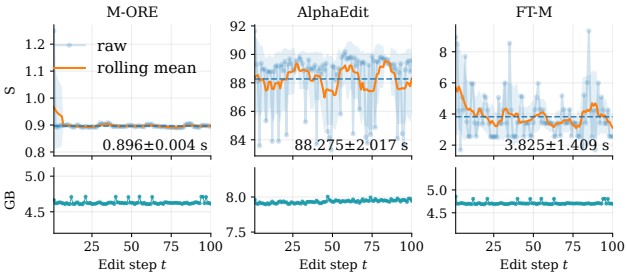

*Figure 7.* Efficiency metrics of editors on LLaVA-v1.5 over the online edit sequence $t$. **Top:** Edit latency (raw and rolling mean) . **Bottom:** Incremental peak memory ($\Delta$ peak mem).

dation observed in other baselines. In contrast, parameter-modifying baselines suffer clear catastrophic forgetting under long horizons: MEND drops to near-zero accuracy on Existence/Count after 100 edits, and FT-M exhibits a steady decline across multiple categories.

### 5.4. Online Editing Efficiency Analysis (RQ3)

We report the *editor-specific* overhead per online edit. Let $r \ll d$ be the steady-subspace rank, $d$ / $d_{\text{out}}$ the FFN key/input and output dimensions, and $\mathcal{L}$ the edited layers. With a constant-size online buffer, M-ORE has $T_{\text{M-ORE}} \approx O\big(|\mathcal{L}|\, d_{\text{out}} r^2\big)$, $M_{\text{M-ORE}} \approx O\big(|\mathcal{L}|\, d_{\text{out}} r\big)$. Both of them are constant with respect to the edit stream length $t$.

Table 5 summarizes per-edit time/memory complexities of representative editors, with derivations provided in Appendix E. M-ORE achieves *constant* per-edit time and memory without relying on dense second-order solvers or edit-growing storage, leading to a favorable long-horizon quality-efficiency trade-off. In contrast, while naive finetuning is cheaper per edit, it fails to preserve multimodal locality under sequential edits (Table 1). Beyond the complexity analysis, we examine the *empirical scaling* of editor-specific overhead along the edit stream. As shown in Figure 7, M-ORE incurs only a short warmup at the first edit (one-off initialization), after which the per-edit latency quickly plateaus and stays stable with increasing $t$, matching the predicted $O(1)$ time. Similarly, $\Delta$peak mem remains flat without

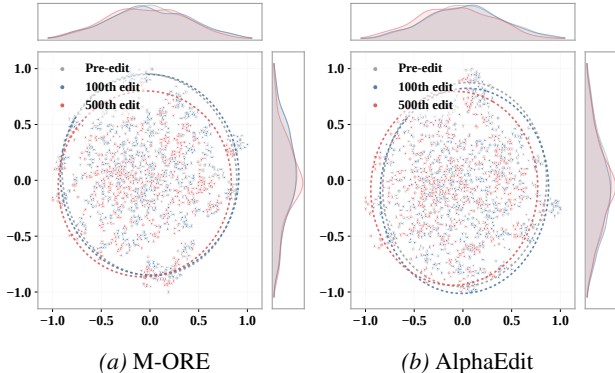

*(a)* M-ORE  *(b)* AlphaEdit

*Figure 8.* Sequential visualization of hidden-state shifts on LLaVA across 500 consecutive locality samples.

monotonic growth, indicating bounded editor state under a constant-size buffer.

### 5.5. Representation Shift Analysis (RQ4)

We evaluate overfitting by measuring the hidden-state distribution shift on locality samples after sequential edits. Since these samples are unrelated to the edited requests, an effective editor should preserve their representations while incorporating new knowledge. As shown in Figure 8, M-ORE keeps the post-edit distributions after the 100th and 500th edits close to the pre-edit distribution, with substantial overlap in both the scatter plots and marginal densities. This indicates that M-ORE does not globally distort the representation space under long edit horizons. AlphaEdit also shows comparable representation stability, but M-ORE achieves this while maintaining stronger long-horizon editing performance in Table 1.

## 6. Conclusion

We studied online editing for multimodal large language models and identified two key obstacles that limit existing editors in practice: *cross-modal conflict* induced by modality-dependent scale mismatch, and *inter-edit interference* caused by long-horizon entanglement under sequential writes. To address both under strict efficiency constraints, we proposed M-ORE, a modality-decoupled online recursive editor derived from a unified proximal-projection view. M-ORE maintains module-wise locality statistics for the text stack and the visual projector, and performs continual updates in a fixed orthogonal low-rank edit subspace via a Sherman-Morrison grounded closed-form recursion, enabling constant per-edit compute and memory. Extensive experiments on representative MLLM backbones and lifelong multimodal editing benchmarks show that M-ORE consistently improves reliability, generality, and locality while preserving general capabilities, achieving a favorable long-horizon quality-efficiency trade-off.

## Impact Statement

This work aims to improve the reliability and maintainability of multimodal large language models by enabling efficient, online updates under strict compute and memory budgets. A practical benefit is that outdated or incorrect multimodal knowledge (e.g., factual errors grounded in images or instructions) can be corrected without expensive retraining, potentially reducing harmful misinformation and improving downstream system robustness.

As with all model editing techniques, the same capability could be misused to inject undesired or misleading behaviors into deployed models. We therefore emphasize the importance of controlled access, auditing, and provenance tracking for edits, as well as evaluating edits under both capability and safety criteria. Our method is designed to preserve pre-edit behaviors via locality constraints, but it does not replace broader safety measures such as content filtering, red-teaming, and policy enforcement.

We release implementation details to support reproducibility and future research on safe and accountable editing, including better monitoring of edit side effects, stronger verification of edit intent, and standardized benchmarks for multimodal online editing.

## Acknowledgements

This work was supported in part by National Natural Science Foundation of China (62476070), Shenzhen Science and Technology Program (JCYJ20241202123503005, GXWD20231128103232001, ZDSYS20230626091203008, KQTD20240729102154066), Department of Science and Technology of Guangdong (2024A1515011540), National Key R&D Program of China (SQ2024YFE0200592), the Major Key Project of PCL under Grant PCL2025A10 and PCL2024A06, and in part by the Shenzhen Science and Technology Program under Grant RCJC20231211085918010.

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

# Appendix

## A. Experimental Setup

In this section, we provide detailed descriptions of experimental setup, including introduction to datasets, explanation of evaluation metrics and editing objective, discussion of baseline methods and implementation details.

### A.1. Datasets

- **E-VQA (Cheng et al., 2023):** Designed for rectifying errors in VQA-v2 (Goyal et al., 2017), this dataset contains 6,346 training and 2,093 testing samples. It requires MLLMs to analyze visual content alongside textual questions to produce precise responses.
- **E-IC (Cheng et al., 2023):** Developed to correct descriptive errors in COCO Caption (Chen et al., 2015), it consists of 2,849 training and 1,000 testing instances. The task demands a comprehensive understanding of images to generate accurate captions.
- **Evaluation Composition:** In addition to the editing samples, the benchmark includes auxiliary samples for evaluating whether edits generalize to equivalent inputs and preserve unrelated knowledge:
    - **Generality:** Textual generality is assessed using semantically rephrased text inputs generated by ChatGLM (Du et al., 2022) for E-VQA and manually written prompt templates for E-IC. Visual generality is assessed using reinterpreted images generated by Stable Diffusion 2.1 (Rombach et al., 2022).
    - **Locality:** Textual locality is evaluated using NQ (Kwiatkowski et al., 2019), and multimodal locality is evaluated using OK-VQA (Marino et al., 2019), measuring whether unrelated knowledge remains unchanged after editing.

### A.2. Evaluation Metrics

We consider an online edit stream $\{e_t = (x_t^v, x_t^q, y_t)\}_{t=1}^T$, where $x_t^v$ is the visual input, $x_t^q$ is the text query, and $y_t$ is the desired edited response. Let $f_{\theta_t}$ denote the model after applying the $t$-th edit. We use $\mathbb{I}(\cdot)$ as the correctness indicator after answer normalization.

**Reliability (Rel.).** Reliability measures whether the edited model produces the desired response on the edited request:

$$\text{Rel} = \frac{1}{T} \sum_{t=1}^T \mathbb{I}(f_{\theta_t}(x_t^v, x_t^q) = y_t). \tag{14}$$

**Generality (T-Gen. / M-Gen.).** Generality evaluates whether the edit generalizes to semantically equivalent variants of the edited request. For each edit step $t$, we define a text-query neighborhood $\mathcal{G}_t^q(x_t^q)$, such as paraphrased queries, and a visual neighborhood $\mathcal{G}_t^v(x_t^v)$, such as benign image transformations or semantically equivalent visual inputs. We report text-side and multimodal-side generality as:

$$\text{T-Gen} = \frac{1}{T} \sum_{t=1}^T \mathbb{E}_{\tilde{x}_t^q \sim \mathcal{G}_t^q(x_t^q)} \left[ \mathbb{I}(f_{\theta_t}(x_t^v, \tilde{x}_t^q) = y_t) \right], \tag{15}$$

$$\text{M-Gen} = \frac{1}{T} \sum_{t=1}^T \mathbb{E}_{\tilde{x}_t^v \sim \mathcal{G}_t^v(x_t^v)} \left[ \mathbb{I}(f_{\theta_t}(\tilde{x}_t^v, x_t^q) = y_t) \right]. \tag{16}$$

**Locality (T-Loc. / M-Loc.).** Locality measures the absence of side effects on inputs unrelated to the current edit. Let $\mathcal{U}_t^q$ denote text-side locality inputs, which are not semantically related to the edited query $x_t^q$, and let $\mathcal{U}_t^v$ denote visual-side locality inputs, which are not semantically related to the edited visual input $x_t^v$. We quantify locality by the distributional shift between the post-edit model $f_{\theta_t}$ and the pre-edit model $f_{\theta_{t-1}}$ using the KL divergence over next-token distributions:

$$\text{T-Loc} = \frac{1}{T} \sum_{t=1}^T \mathbb{E}_{\tilde{x}_t^q \sim \mathcal{U}_t^q} \left[ \exp\left(-\text{KL}\left(p_{\theta_t}(\cdot | \tilde{x}_t^q) \, \| \, p_{\theta_{t-1}}(\cdot | \tilde{x}_t^q)\right)\right) \right], \tag{17}$$

$$\text{M-Loc} = \frac{1}{T} \sum_{t=1}^T \mathbb{E}_{(\tilde{x}_t^v, \tilde{x}_t^q) \sim \mathcal{U}_t^v} \left[ \exp\left(-\text{KL}\left(p_{\theta_t}(\cdot | \tilde{x}_t^v, \tilde{x}_t^q) \, \| \, p_{\theta_{t-1}}(\cdot | \tilde{x}_t^v, \tilde{x}_t^q)\right)\right) \right]. \tag{18}$$

Here $p_\theta(\cdot \mid x)$ is the model's next-token distribution under the same decoding prefix, and higher values indicate better locality.

### A.3. Edit objective $\mathcal{L}_{\text{edit}}$.

At edit step $t$, we optimize the negative log-likelihood of the desired edited responses:

$$\mathcal{L}_{\text{edit}}^t = \mathbb{E}_{(x_t^v, x_t^q, y_t)} \Big[ - \log p_\theta \big( y_t \mid x_t^v, x_t^q \big) \Big]. \tag{19}$$

### A.4. MLLM Backbones

We employ several representative Multimodal Large Language Models (MLLMs) as our experimental backbones. Specific model versions are detailed in the following.

- **LLaVA**[2] (**Liu et al., 2024**): LLaVA aligns vision and language by using an MLP-based visual projector to map image features into the embedding space of LLaMA (Liu et al., 2024). With GPT-4-generated instruction-following data (OpenAI, 2023), it shows strong fine-grained visual reasoning and complex instruction following.
- **BLIP-2**[3] (**Li et al., 2023**): BLIP-2 introduces a lightweight Q-Former and a two-stage pre-training pipeline to bridge a frozen vision encoder and a frozen LLM. Following (Cheng et al., 2023; Chen et al., 2025a), we adopt the BLIP2-OPT variant, which prioritizes inference efficiency by compressing visual tokens before interfacing with OPT.

### A.5. Baseline Editors

**Fine-tuning (FT).**  We include two finetuning variants commonly used for MLLM editing: FT-L and FT-M (Cheng et al., 2023). FT-L updates (only) the last layer of the language transformer, while FT-M finetunes the visual encoder (or vision-side adaptation module) for each edit sample, following the setup in prior work (Cheng et al., 2023).

**MEND.**  Model Editor Networks with Decomposition (Mitchell et al., 2022a) is a hypernetwork-based editor that learns to predict parameter updates efficiently. It trains a set of lightweight MLP hypernetworks that take decomposed backpropagated gradients on edit samples as input and output offsets to the target FFN parameters. After editor-specific training, these hypernetworks can generate per-edit updates that satisfy the editing objective with relatively low runtime overhead.

**AlphaEdit.**  AlphaEdit (Fang et al., 2025) is a plug-and-play, optimization-based editor that achieves targeted updates while explicitly preserving pre-edit behaviors. Under a locate-then-edit pipeline, it first solves for a task-specific weight update on a chosen subset (e.g., FFN $W_{\text{out}}$), and then projects this update into the null space induced by a set of *retain* keys, enforcing invariance on those activations to control side effects. This null-space projection also helps reduce interference across sequential edits by keeping previously protected mappings unchanged.

**SERAC.**  Semi-parametric Editing with a Retrieval-Augmented Counterfactual (Mitchell et al., 2022b) is a memory-based editing method. It stores edited samples and trains (i) a *scope classifier* to detect whether a query is related to previous edits, and (ii) a small *counterfactual model* to produce the modified response when the query falls within the edit scope. Otherwise, the original model is used for generation. Following the standard setup (Chen et al., 2025a), we instantiate the scope classifier with BERT (Devlin et al., 2019) and the counterfactual model with OPT-125M (Zhang et al., 2022).

**IKE.**  In-Context Knowledge Editing (Zheng et al., 2023) performs editing by *retrieval-augmented in-context prompting*, without directly updating model parameters. Given a target fact pair $(x^*, y^*)$, IKE retrieves $k$ demonstrations $C = \{c_1, \ldots, c_k\}$ from a training set using an unsupervised retriever (e.g., cosine similarity), and concatenates them as in-context examples to guide generation. The demonstrations are ordered by similarity to the target, and the resulting augmented prompt aims to maximize $P(y \mid x, f, C)$ for inputs $x$ that fall within the scope of the target prompt.

**LiveEdit.**  LiveEdit (Chen et al., 2025a) is designed for *lifelong / streaming* edits in vision–language models, aiming to maintain edit quality over long horizons. Instead of repeatedly overwriting shared weights, it stores edits as lightweight low-rank experts (a low-rank MoE) and performs gated composition at inference, with routing that favors visually and

---

[2] https://huggingface.co/liuhaotian/llava-v1.5-7b
[3] https://huggingface.co/Salesforce/blip2-opt-2.7b

textually relevant experts to mitigate inter-edit interference. These mechanisms promote stability and reduce cumulative drift under continuous updates.

### A.6. Implementation Details

**Aligned baseline protocols.** For FT-L, FT-M, MEND, SERAC, and LiveEdit, we follow and align with the official editing protocol in LiveEdit/MMEdit (Chen et al., 2025a; Cheng et al., 2023), including the same edit-stream construction, training/evaluation splits, and per-edit optimization settings. For IKE, we use the MMEdit-aligned setup (Cheng et al., 2023) to ensure a fair comparison under identical edit scopes.

**AlphaEdit configuration and multimodal $K_0$.** For AlphaEdit, we adopt the hyperparameters recommended in the original paper (Fang et al., 2025). To estimate the retain key set $K_0$ for MLLMs, we build $K_0$ using samples from E-VQA and E-IC (Cheng et al., 2023), so that both visual and textual knowledge are covered when constructing the null-space constraint. We restrict the editable modules to the last seven transformer layers of each MLLM, since these upper layers are the primary contribution layers identified in Section 3.2.

**M-ORE configuration.** We perform online updates with M-ORE using the closed-form solver (Eq. (12)), without iterative optimization. To make the setting more challenging, we use a per-edit batch size of 1 throughout. We apply model-specific hyperparameters as follows:

- **LLaVA-v1.5.** We set the LoRA rank to $r = 512$ with $\alpha/r = 2.0$. We use shared hyperparameters for the language and vision components: $\eta = 0.1$ and $\lambda = 2000$. We update the last seven transformer layers as well as the visual projection layer. We initialize $A^{(l)}$ using `torch.nn.init.orthogonal`. The online-updated matrix $B^{(l)}$ is initialized to zeros, ensuring the pre-edit model remains unchanged at step 0.
- **BLIP2-OPT.** We set the LoRA rank to $r = 128$ with $\alpha/r = 2.0$. For the language backbone, we use $\eta = 0.06$ and $\lambda = 5000$; for the visual projection, we use $\eta_{\text{vis}} = 0.03$ and $\lambda_{\text{vis}} = 20000$. We update the last seven transformer layers as well as the visual projection layer. We use the same initialization strategy as above.

**Codebase.** For fairness and reproducibility, we implement all methods (including M-ORE and baselines) on top of the widely-used EASYEDIT editing framework[4], and make our modifications within its unified editing/evaluation pipeline. All experiments are conducted on a single NVIDIA H20 GPU (96GB) with NVLink. We additionally provide our M-ORE implementation in the https://github.com/lab-klc/M-ORE

## B. Supplementary Experiments

### B.1. Cross-modal Conflict

For the attribution analysis, we sample four instances from E-IC and evaluate LLaVA-v1.5 and BLIP2-OPT. We quantify layer-wise attributions for the self-attention, MLP, and visual projector modules by aggregating their activation magnitudes across the four representative samples to derive the final contribution profiles. We additionally compute diagonal activation variances (log-variance) for textual layers and the projected visual representations. We provide the corresponding cross-modal conflict results for LLaVA-v1.5 in Figure 9.

To further verify whether the identified cross-modal conflict persists in more recent MLLM architectures, we additionally examine Qwen2-VL-7B-Instruct (Wang et al., 2024a) [5] and Qwen3-VL-8B-Instruct (Bai et al., 2025) [6]. As shown in Table 6, both models exhibit a clear modality-scale mismatch: the visual projection layer produces substantially larger mean outputs than representative MLP layers. Specifically, the visual projection output reaches 17.67 on Qwen2-VL and 51.51 on Qwen3-VL, while the representative MLP-layer outputs are only in the ranges of 4.60–11.66 and 10.03–25.50, respectively. These results indicate that the cross-modal conflict is not specific to BLIP2-OPT or LLaVA-v1.5, but remains observable in newer MLLM backbones.

---

[4] https://github.com/zjunlp/EasyEdit
[5] https://huggingface.co/Qwen/Qwen2-VL-7B-Instruct
[6] https://huggingface.co/Qwen/Qwen3-VL-8B-Instruct

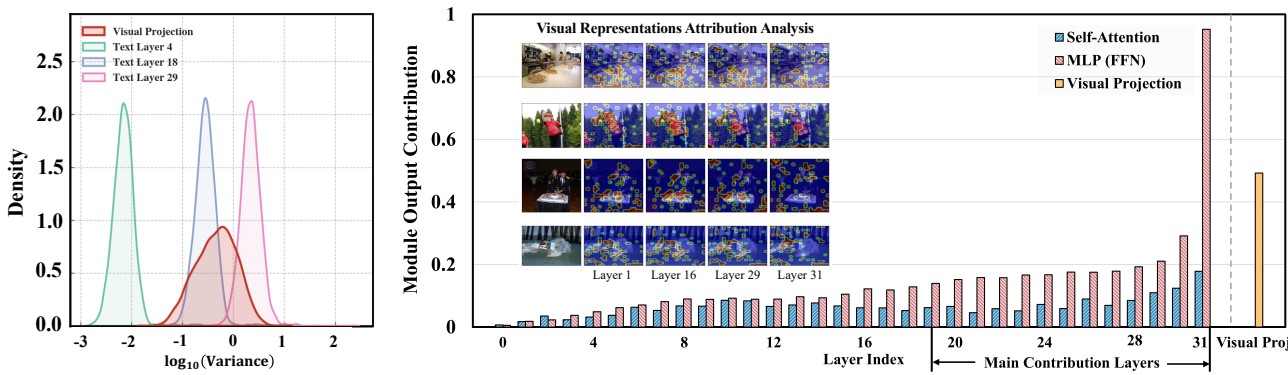

*Figure 9.* Cross-modal conflict caused by modality mismatch (LLaVA-v1.5). **Left:** log-variance density indicates higher-energy visual features than text activations. **Right:** attribution analysis shows visually dominated contributions, implying that shared global statistics bias updates toward the visual subspace and weaken textual preservation.

*Table 6.* Modality-scale mismatch on recent MLLM backbones. We report the mean output magnitudes of the visual projection layer and representative MLP layers.

| Backbone | Visual Projection | Representative MLP Layers |
|----------|-------------------|---------------------------|
| Qwen2-VL | 17.67 | 4.60–11.66 |
| Qwen3-VL | 51.51 | 10.03–25.50 |

## B.2. Comprehensive Performance Comparison (RQ1)

We provide the complete online editing results on E-VQA and E-IC for BLIP2-OPT and LLaVA-v1.5 under all edit horizons in Table 11, which supplements the partial results reported in the main paper.

## B.3. General Capability Evaluation (RQ2)

MME (Fu et al., 2023) contains 14 evaluation categories. The main paper reports six representative tasks for brevity, while we provide the remaining eight categories here to complete the benchmark:

- **Artwork**: evaluates understanding of artistic images and stylized visual content.
- **Celebrity**: tests recognition and reasoning about well-known public figures in images.
- **Color**: measures color perception and discrimination for queried objects/regions.
- **Landmark**: evaluates recognition of landmarks and related visual reasoning.
- **Poster**: assesses understanding of poster-like images with dense visual-textual layouts.
- **Scene**: tests holistic scene understanding and context-aware reasoning.
- **Text Translation**: evaluates translating text appearing in images into the target language.
- **Code Reasoning**: probes reasoning over code-like or structured text content presented visually.

Figure 10 summarizes the accuracy trajectories of post-edited LLaVA-v1.5 across these additional tasks under the same evaluation protocol.

## B.4. Hyperparameter Analysis

**Sensitivity to rank $r$.** Figure 11 studies the LoRA rank $r \in \{128, 256, 512, 1024\}$. Increasing $r$ improves edit capacity and generally benefits multimodal generalization under long horizons, while the gains saturate beyond a moderate rank (e.g., $r=512$). Small ranks tend to underfit multimodal updates, consistent with insufficient write capacity.

**Sensitivity to locality regularization $\lambda$.** Figure 12 varies $\lambda$ in $\{100, 1000, 2000, 3000\}$. Small $\lambda$ leads to worse locality and lower average scores, indicating insufficient preservation of pre-edit behavior, whereas strong regularization can slightly reduce plasticity. A moderate range yields the best overall stability-plasticity trade-off across edit horizons.

*Table 7.* Online E-IC editing results of M-ORE on recent other MLLM backbones.

| Backbone | #Edit | Rel. | T-Gen. | M-Gen. | T-Loc. | M-Loc. | Avg. |
|----------|-------|------|--------|--------|--------|--------|------|
| Qwen2-VL | 1   | 100.00 | 100.00 | 92.31 | 100.00 | 96.67 | 97.80 |
| Qwen2-VL | 10  | 96.56 | 97.33 | 85.18 | 91.47 | 90.67 | 92.24 |
| Qwen2-VL | 100 | 93.66 | 93.51 | 80.15 | 89.43 | 90.75 | 89.50 |
| Qwen3-VL | 1   | 100.00 | 100.00 | 84.62 | 100.00 | 100.00 | 96.92 |
| Qwen3-VL | 10  | 95.51 | 96.35 | 88.84 | 95.93 | 97.55 | 94.84 |
| Qwen3-VL | 100 | 91.85 | 92.54 | 80.50 | 91.63 | 93.33 | 89.97 |

### B.5. Extension to More Intricate Vision-Language Alignment Architectures

First, we evaluate online E-IC editing on more recent MLLM backbones, including Qwen2-VL-7B-Instruct and Qwen3-VL-8B-Instruct. Importantly, we directly reuse the hyperparameter setting from LLaVA-v1.5 without architecture-specific retuning. As shown in Table 7, M-ORE maintains strong performance across different edit horizons on both newer architectures. Even after $t=100$ edits, M-ORE achieves average scores of 89.50 on Qwen2-VL and 89.97 on Qwen3-VL, indicating that its online recursive editing mechanism remains effective across modern MLLM architectures.

Second, M-ORE does not assume that the visual projector is a single homogeneous module. Its decoupling principle is module-wise: when a backbone contains multiple vision-language alignment modules, M-ORE can assign an independent locality statistic and preconditioner to each edited module. This confines high-energy visual features to the corresponding vision-side statistics and prevents them from contaminating the update geometry of text-side modules. To verify this extension beyond the default projector-plus-text-LLM setting, we further evaluate vision-side parameter editing on LLaVA-v1.5 under $t=100$ edits. Besides the default setting that edits the projector and text-LLM modules, we consider two variants: **(1) editing the vision encoder and projector only**, and (**2) editing the vision encoder, projector, text-LLM jointly**. For the vision encoder, we edit only its last three layers.

As shown in Tables 8 and 9, M-ORE remains compatible with vision-side editing. Editing the vision encoder and projector alone preserves high text locality but yields lower reliability and generality, suggesting that vision-side updates alone are insufficient to fully absorb multimodal factual corrections. Jointly editing vision-side modules and text-LLM modules improves multimodal generality, especially on E-VQA, but slightly reduces the overall average compared with the default setting. These results support our design choice of applying module-wise decoupled statistics to the projector and text-LLM modules by default, while also showing that the same principle naturally extends to more intricate alignment architectures with multiple editable vision-language modules.

### B.6. Case Study

We provide qualitative case studies on both E-IC and E-VQA using LLaVA-v1.5, covering challenging edits that require fine-grained visual grounding (Figures 13–16). For each example, we report the target concept, the original prompt, the model's outputs before/after editing, and a paraphrased prompt to probe text generality. To visualize grounding and potential side effects, we additionally plot image-space attribution/attention rollout maps for the pre-edit model (`Base`) and the post-edit model at representative layers (`L1`, `L16`, `L31`). Each group contains an edited sample (top) and an unrelated locality sample (bottom). We summarize the main qualitative findings as follows:

- **Edit success & paraphrase robustness.** M-ORE consistently corrects the edited samples to the target responses, and the correction remains valid under paraphrased prompts.
- **Locality preservation.** On the paired locality samples, the post-edit model largely preserves the pre-edit predictions, indicating limited side effects.
- **Grounded attribution shifts.** Attribution maps show that, after editing, deeper layers place more mass on image regions supporting the updated concept. In contrast, locality samples exhibit spatial patterns close to the pre-edit baseline, suggesting the edit is grounded rather than inducing spurious global evidence shifts.
- **Occasional locality flips.** In a small number of locality cases, the decoded output changes after editing. To better understand the cause of these flips, we compare stable and flipped locality cases in Table 10. As shown, flipped cases tend to have smaller pre-edit top1–top2 margins, indicating that *they lie near fragile decoding boundaries and can be affected by small distributional shifts*. For multimodal locality, flipped cases also show higher image similarity to the edited sample, suggesting that *imperfect visual irrelevance and residual visual coupling may contribute to occasional flips*. In contrast, the low prompt token-F1 suggests that these flips are not primarily caused by surface-level textual overlap.

*Table 8.* Vision-side extension results on E-VQA with LLaVA-v1.5 under $t{=}100$ online edits.

| Variant | Rel. | T-Gen. | M-Gen. | T-Loc. | M-Loc. | Avg. |
|---|---|---|---|---|---|---|
| **M-ORE (default: projector + text-LLM)** | **97.43** | **94.30** | 85.40 | 96.32 | **91.90** | **93.07** |
| M-ORE w/ vis (vision encoder + projector only) | 83.16 | 80.42 | 77.77 | **100.00** | 79.80 | 84.23 |
| M-ORE w/ vis-text (vision encoder + projector + text-LLM) | 92.32 | 87.18 | **92.82** | 93.92 | 87.17 | 90.68 |

*Table 9.* Vision-side extension results on E-IC with LLaVA-v1.5 under $t{=}100$ online edits.

| Variant | Rel. | T-Gen. | M-Gen. | T-Loc. | M-Loc. | Avg. |
|---|---|---|---|---|---|---|
| **M-ORE (default: projector + text-LLM)** | **94.64** | **93.27** | 78.66 | 97.39 | **89.71** | **90.73** |
| M-ORE w/ vis (vision encoder + projector only) | 81.67 | 81.34 | 62.44 | **100.00** | 81.50 | 81.39 |
| M-ORE w/ vis-text (vision encoder + projector + text-LLM) | 91.07 | 90.33 | **80.98** | 98.05 | 87.95 | 89.68 |

*Table 10.* Comparison between stable and flipped locality cases. Pre top1–top2 margin measures pre-edit decoding confidence; mean gold shift measures the distributional movement toward the locality reference answer; prompt token-F1 measures lexical overlap between the edit prompt and locality prompt; image cosine measures visual similarity between the edit image and multimodal locality image.

| Metric | Text stable | Text flip | MM stable | MM flip |
|---|---|---|---|---|
| Pre top1–top2 margin | 3.28 | **2.46** | 2.50 | **1.07** |
| Mean gold shift | 0.0053 | 0.0040 | 0.0002 | **0.0901** |
| Prompt token-F1 | 0.077 | 0.072 | 0.063 | **0.031** |
| Image cosine | – | – | 0.062 | **0.319** |

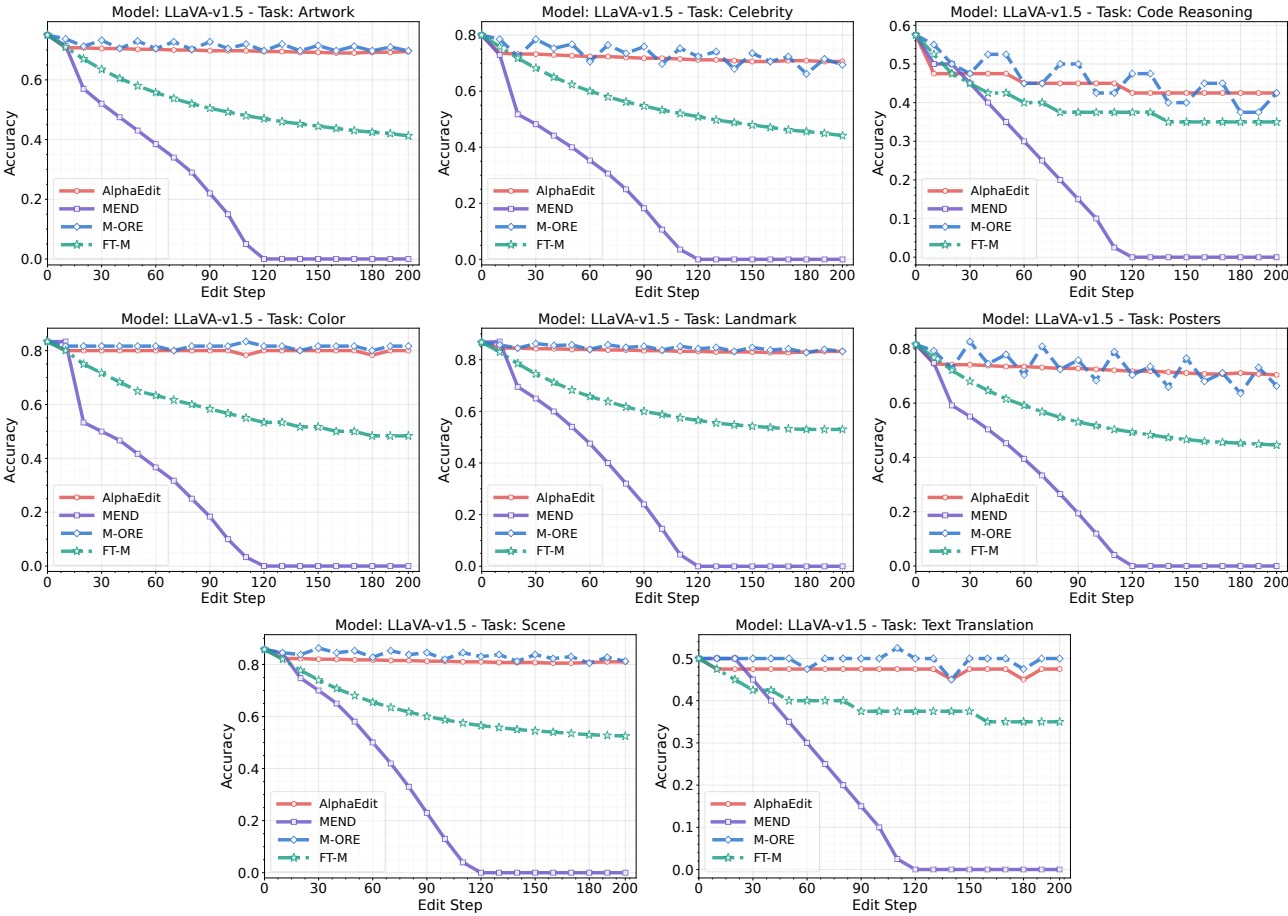

*Figure 10.* Accuracy of the post-edited LLaVA-v1.5 (7B) on remaining eight tasks used for MLLM general capability testing.

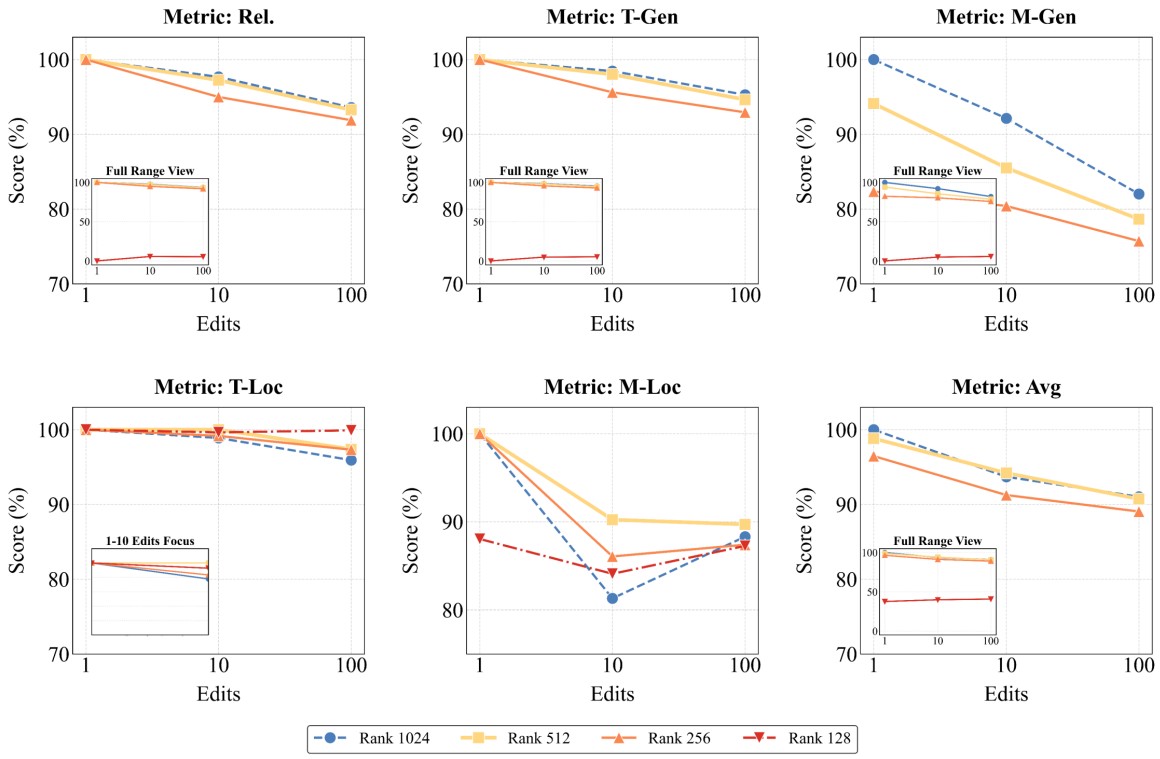

*Figure 11.* Sensitivity analysis of the LoRA rank.

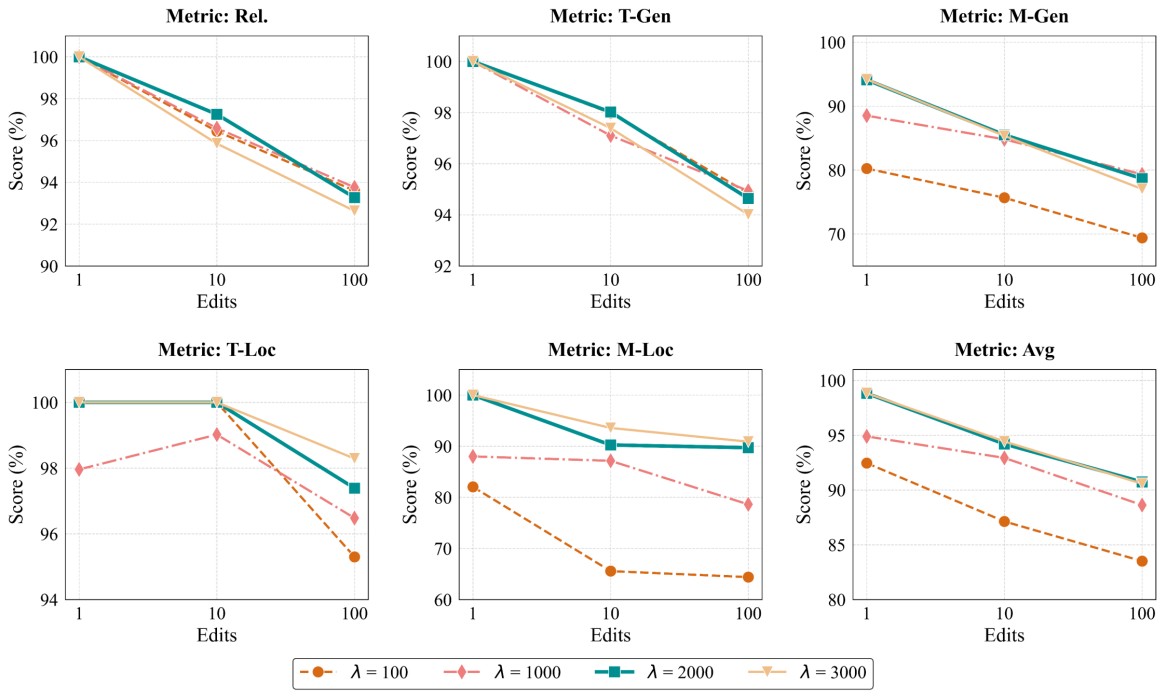

*Figure 12.* Sensitivity analysis of the regularization weight $\lambda$.

## C. Relation to Locate-then-Editing Editors

We relate our proximal-projection update (Eq. (5)) to the classical *target-matching* formulation that enforces $(W + \Delta W)K \approx V$, where $K$ and $V$ stack the key and value vectors, respectively. We establish a first-order connection for the quadratic objective and explain why our formulation is better suited to online sequential editing.

**Target matching.** Let $W \in \mathbb{R}^{d_{\text{out}} \times d}$, keys $K \in \mathbb{R}^{d \times n}$, and desired values $V \in \mathbb{R}^{d_{\text{out}} \times n}$. A standard objective is

$$\Delta W_{\text{tm}} = \arg\min_{\Delta W} \; \|(W + \Delta W)K - V\|_F^2 + \|\Delta W\|_F^2. \tag{20}$$

**Proximal view.** Define $\mathcal{L}_{\text{tm}}(W) = \|WK - V\|_F^2$ and let

$$g \triangleq \nabla_W \mathcal{L}_{\text{tm}}(W) = 2(WK - V)K^\top, \tag{21}$$

where $\langle A, B \rangle \triangleq \text{Tr}(A^\top B)$ denotes the Frobenius inner product. A gradient-centered proximal step can be viewed as a special case of Eq. (5) with an isotropic locality geometry $C = I$:

$$\Delta W_{\text{prox}} = \arg\min_{\Delta W} \; \|\Delta W + \eta g\|_F^2 + \|\Delta W\|_F^2 = -\frac{\eta}{2} g. \tag{22}$$

**Lemma C.1** (First-order descent and preconditioned form). *Assume $\eta > 0$. For the quadratic loss $\mathcal{L}_{tm}$, both $\Delta W_{prox}$ and $\Delta W_{tm}$ are first-order descent directions, i.e., $\langle g, \Delta W_{prox} \rangle < 0$ and $\langle g, \Delta W_{tm} \rangle < 0$ whenever $g \neq 0$. Moreover, $\Delta W_{tm}$ admits the closed form*

$$\Delta W_{tm} = -\tfrac{1}{2} g \left(KK^\top + I\right)^{-1}, \tag{23}$$

*hence it is a* right-preconditioned *gradient step. In particular, if $KK^\top \approx cI$ (e.g., under approximately whitened keys), then $\Delta W_{tm}$ is approximately proportional to g.*

*Proof.* For any perturbation $\Delta W$, a first-order Taylor expansion gives

$$\mathcal{L}_{\text{tm}}(W + \Delta W) = \mathcal{L}_{\text{tm}}(W) + \langle g, \Delta W \rangle + O(\|\Delta W\|_F^2).$$

From Eq. (22), $\Delta W_{\text{prox}} = -\alpha g$ with $\alpha = \eta/2 > 0$, so

$$\langle g, \Delta W_{\text{prox}} \rangle = -\alpha \|g\|_F^2 < 0 \quad \text{when } g \neq 0.$$

For Eq. (20), the first-order optimality condition is

$$2\big((W + \Delta W_{\text{tm}})K - V\big)K^\top + 2\Delta W_{\text{tm}} = 0,$$

which rearranges to

$$\Delta W_{\text{tm}}(KK^\top + I) = (V - WK)K^\top, \;\; \Delta W_{\text{tm}} = -(WK - V)K^\top(KK^\top + I)^{-1}.$$

Since $(KK^\top + I) \succ 0$ is invertible, this yields Eq. (23). Finally,

$$\langle g, \Delta W_{\text{tm}} \rangle = -\tfrac{1}{2} \text{Tr}\big(g^\top g(KK^\top + I)^{-1}\big) < 0 \quad \text{when } g \neq 0.$$

Since $(KK^\top + I)^{-1} \succ 0$, the proportionality remark follows from $(KK^\top + I)^{-1} \approx \frac{1}{c+1}I$ when $KK^\top \approx cI$. $\qquad\square$

**Proposition C.2** (Online suitability). *At edit step $t$, Eq. (5) explicitly imposes a history-dependent quadratic locality geometry $\text{Tr}(\Delta W \, C_{t-1} \, \Delta W^\top)$, where $C_{t-1}$ contains only statistics accumulated from previous edit steps. After the current edit, the step-only context is inserted into the statistic to obtain $C_t$ for future edits. This makes the update naturally suited to online sequential editing: the current write is protected by past locality information, while the current context is cached only for subsequent steps. In contrast, a one-shot target-matching objective of the form Eq. (20) does not encode this cross-edit geometry unless one augments it with additional preservation terms, e.g., a $C_{t-1}$-weighted quadratic penalty at step $t$. In that case, the resulting update becomes equivalent in spirit to our proximal-projection formulation.*

## D. Derivations for Proximal Projection and Steady-Space RLS

We consider Eq. (5) at edit step $t$:

$$\min_{\Delta W} \|\Delta W + \eta g_t\|_F^2 + \mathrm{Tr}(\Delta W C_{t-1} \Delta W^\top), \quad C_{t-1} \succeq 0.$$

Using $\nabla_{\Delta W} \|\Delta W + \eta g_t\|_F^2 = 2(\Delta W + \eta g_t)$ and, since $C_{t-1}$ is symmetric PSD ($C_t \succeq 0$),

$$\nabla_{\Delta W} \mathrm{Tr}(\Delta W C_{t-1} \Delta W^\top) = 2\Delta W C_{t-1},$$

setting the gradient to zero yields

$$(\Delta W + \eta g_t) + \Delta W C_{t-1} = 0 \;\Rightarrow\; \Delta W(I + C_{t-1}) = -\eta g_t \;\Rightarrow\; \Delta W_t = -\eta \, g_t (I + C_{t-1})^{-1}.$$

Since $I + C_{t-1} \succ 0$, the minimizer is unique.

### D.1. From Full-Space Locality to the Steady Orthogonal Coordinates

Under the low-rank interface $\Delta W^{(l)} = \Delta B^{(l)} A^{(l)}$, the locality quadratic form used at edit step $t$ transforms as

$$\mathrm{Tr}\Big(\Delta W^{(l)} C_{t-1}^{(l)} (\Delta W^{(l)})^\top\Big) = \mathrm{Tr}\Big(\Delta B^{(l)} A^{(l)} C_{t-1}^{(l)} A^{(l)^\top} (\Delta B^{(l)})^\top\Big)$$
$$= \mathrm{Tr}\Big(\Delta B^{(l)} \widetilde{C}_{t-1}^{(l)} (\Delta B^{(l)})^\top\Big),$$

where

$$\widetilde{C}_{t-1}^{(l)} \triangleq A^{(l)} C_{t-1}^{(l)} A^{(l)^\top} \in \mathbb{R}^{r \times r}.$$

Moreover, with $z_s^{(l)}(x) = A^{(l)} k_s^{(l)}(x)$ and

$$C_{t-1}^{(l)} = \sum_{s=1}^{t-1} \sum_{x \in \mathcal{B}_s^{(l)}} k_s^{(l)}(x) k_s^{(l)}(x)^\top,$$

we have

$$\widetilde{C}_{t-1}^{(l)} = \sum_{s=1}^{t-1} \sum_{x \in \mathcal{B}_s^{(l)}} z_s^{(l)}(x) z_s^{(l)}(x)^\top.$$

To keep constant per-edit overhead, we approximate each step-wise second-order statistic by a rank-one sketch using the pooled feature $\bar{z}_s^{(l)}$ and maintain $S_{t-1}^{(l)}$ via Eq. (10).

### D.2. Closed Form and Sherman–Morrison Recursion in the Steady Space

At edit step $t$, the current write uses the steady-space statistic accumulated before the current step. Consider Eq. (11):

$$\min_{\Delta B^{(l)}} \|\Delta B^{(l)} + \eta G_t^{(l)}\|_F^2 + \mathrm{Tr}\Big(\Delta B^{(l)} S_{t-1}^{(l)} (\Delta B^{(l)})^\top\Big).$$

Taking derivatives gives

$$2(\Delta B^{(l)} + \eta G_t^{(l)}) + 2\Delta B^{(l)} S_{t-1}^{(l)} = 0,$$

hence

$$\Delta B_t^{(l)} = -\eta \, G_t^{(l)} (I + S_{t-1}^{(l)})^{-1}.$$

Define

$$P_{t-1}^{(l)} \triangleq (I + S_{t-1}^{(l)})^{-1}.$$

Then the current write is

$$\Delta B_t^{(l)} = -\eta \, G_t^{(l)} P_{t-1}^{(l)}.$$

After applying the current write, the current step feature $\bar{z}_t^{(l)}$ is inserted into the recursive statistic:

$$S_t^{(l)} = S_{t-1}^{(l)} + \bar{z}_t^{(l)} (\bar{z}_t^{(l)})^\top.$$

Therefore,

$$(I + S_t^{(l)}) = (I + S_{t-1}^{(l)}) + \bar{z}_t^{(l)} (\bar{z}_t^{(l)})^\top.$$

Applying the Sherman–Morrison lemma with $u = v = \bar{z}_t^{(l)}$ yields

$$P_t^{(l)} = P_{t-1}^{(l)} - \frac{P_{t-1}^{(l)} \bar{z}_t^{(l)} (\bar{z}_t^{(l)})^\top P_{t-1}^{(l)}}{1 + (\bar{z}_t^{(l)})^\top P_{t-1}^{(l)} \bar{z}_t^{(l)}},$$

which is Eq. (13). Thus, $P_{t-1}^{(l)}$ is used to compute the current write $\Delta B_t^{(l)}$, while the updated $P_t^{(l)}$ is cached for the next edit step $t + 1$. Finally, because $S_0^{(l)} = \lambda I_r$ with $\lambda > 0$ and each update adds a PSD rank-one matrix, $I + S_t^{(l)} \succ 0$ for all $t$. Hence $P_t^{(l)}$ is always well-defined. The recursion updates each layer in $O(r^2)$ time and stores only $P_t^{(l)} \in \mathbb{R}^{r \times r}$.

## E. Complexity Derivations and Details

**Scope and accounting.** We analyze the *editor-specific* overhead per online edit step, i.e., (i) statistics maintained by the editor and (ii) parameter updates applied by the editor. We *exclude* the forward/backward cost for computing edit gradients, since it is shared by parameter-modifying editors under the same training objective and hardware settings. Unless stated otherwise, we assume a constant-size step-only buffer $\mathcal{B}_t$ (per layer) with $b \triangleq |\mathcal{B}_t| = O(1)$.

### E.1. Notation

Let $l \in \mathcal{L}$ index the edited FFN layers. Denote the FFN key/input dimension by $d$ and the FFN output dimension by $d_{\text{out}}$. M-ORE uses a fixed steady-space rank $r \ll d$ and a frozen orthogonal basis $A^{(l)} \in \mathbb{R}^{r \times d}$ (Eq. (7) in the main paper). The editable write parameters are $B_{t-1}^{(l)} \in \mathbb{R}^{d_{\text{out}} \times r}$. At edit step $t$, let $k_t^{(l)}(x) \in \mathbb{R}^d$ be the FFN key vector for editing sample input $x$, and $G_t^{(l)} = \nabla_{B_{t-1}^{(l)}} \mathcal{L}_{\text{edit}} \in \mathbb{R}^{d_{\text{out}} \times r}$ the edit gradient in the steady space. The steady-space statistic is $\bar{z}_t^{(l)} \in \mathbb{R}^r$ and the preconditioner is $P_t^{(l)} \in \mathbb{R}^{r \times r}$.

### E.2. M-ORE: Per-edit Time Complexity

**Step A: preconditioned write.** The layer-wise write at edit step $t$ uses the preconditioner accumulated before the current step:

$$\Delta B_t^{(l)} = -\eta\, G_t^{(l)} P_{t-1}^{(l)}, \tag{24}$$

where $G_t^{(l)} \in \mathbb{R}^{d_{\text{out}} \times r}$ and $P_{t-1}^{(l)} \in \mathbb{R}^{r \times r}$. Right-multiplying by $P_{t-1}^{(l)}$ costs $O(d_{\text{out}} r^2)$.

**Step B: pooled steady-space statistic.** After the current write, M-ORE forms the pooled key for updating the next preconditioner:

$$\bar{k}_t^{(l)} \triangleq \text{Mean}_{x \in \mathcal{B}_t^{(l)}} \big(k_t^{(l)}(x)\big) \in \mathbb{R}^d, \tag{25}$$

which requires a reduction over $b$ vectors of length $d$, i.e., $O(bd)$ time. It then projects the pooled key into the steady subspace,

$$\bar{z}_t^{(l)} = A^{(l)} \bar{k}_t^{(l)} \in \mathbb{R}^r, \tag{26}$$

which costs $O(rd)$ time since $A^{(l)} \in \mathbb{R}^{r \times d}$. Equivalently, by linearity, $\bar{z}_t^{(l)} = \text{Mean}_{x \in \mathcal{B}_t^{(l)}} \big(A^{(l)} k_t^{(l)}(x)\big)$.

**Step C: Sherman–Morrison update for the next preconditioner.** M-ORE inserts the current step feature $\bar{z}_t^{(l)}$ into the recursive statistic via the Sherman–Morrison lemma, yielding the preconditioner for the next edit step:

$$P_t^{(l)} = P_{t-1}^{(l)} - \frac{P_{t-1}^{(l)} \bar{z}_t^{(l)} (\bar{z}_t^{(l)})^\top P_{t-1}^{(l)}}{1 + (\bar{z}_t^{(l)})^\top P_{t-1}^{(l)} \bar{z}_t^{(l)}}. \tag{27}$$

This can be computed via the standard rank-one routine: compute $u = P_{t-1}^{(l)} \bar{z}_t^{(l)}$ in $O(r^2)$ time, compute the scalar denominator in $O(r)$ time, and apply the outer-product update $P_{t-1}^{(l)} - uu^\top / \text{den}$ in $O(r^2)$ time. Thus, Step C is $O(r^2)$.

**Total per-edit time (exact and simplified).**  Summing Steps A–C for a single layer gives

$$O\big(d_{\text{out}}r^2 + bd + rd + r^2\big). \tag{28}$$

Aggregating over all edited layers $|\mathcal{L}|$ yields the exact per-edit time:

$$T_{\text{M-ORE}} = O\Big(|\mathcal{L}|\,(d_{\text{out}}r^2 + bd + rd + r^2)\Big). \tag{29}$$

With a constant-size buffer ($b = O(1)$) and $r \ll d, d_{\text{out}}$, we drop lower-order terms and obtain:

$$T_{\text{M-ORE}} = O\Big(|\mathcal{L}|\,(d_{\text{out}}r^2 + rd + r^2)\Big) \approx O\big(|\mathcal{L}|\,d_{\text{out}}r^2\big). \tag{30}$$

### E.3. M-ORE: Per-edit Space Complexity

**Stored state.**  For each edited layer $l \in \mathcal{L}$, M-ORE stores: (i) the frozen low-rank write basis $A^{(l)} \in \mathbb{R}^{r \times d}$, (ii) the low-rank write parameters $B^{(l)} \in \mathbb{R}^{d_{\text{out}} \times r}$, and (iii) the steady-space preconditioner $P^{(l)} \in \mathbb{R}^{r \times r}$. Therefore, the memory cost is

$$M_{\text{M-ORE}} = O\Big(|\mathcal{L}|\,(rd + d_{\text{out}}r + r^2)\Big). \tag{31}$$

Since $r \ll d, d_{\text{out}}$ and $d$ is typically of the same order as $d_{\text{out}}$ for the edited modules, the lower-order $r^2$ term can be ignored, yielding

$$M_{\text{M-ORE}} \approx O\Big(|\mathcal{L}|\,r(d + d_{\text{out}})\Big) \approx O\big(|\mathcal{L}|\,d_{\text{out}}r\big). \tag{32}$$

The buffer memory is $O(bd)$ per layer if keys are stored explicitly; in our implementation $b$ is constant and keys can be recomputed on-the-fly, so the asymptotic dependence on the edit length $t$ remains unchanged.

**Independence from edit length $t$.**  M-ORE maintains only fixed-size per-layer state ($B^{(l)}$ and $P^{(l)}$) and does not store edit-specific experts or retrieval items. Thus, both $T_{\text{M-ORE}}$ and $M_{\text{M-ORE}}$ are $O(1)$ with respect to the edit stream length $t$.

### E.4. Baseline Complexity Derivations

Below, we justify the baseline entries in Table 5. We again consider editor-specific overhead; when a method also induces inference-time costs (e.g., selection/retrieval), we report those costs explicitly.

#### E.4.1. LOCATE-THEN-EDIT

Many locate-then-edit editors require solving a regularized system involving a dense second-order statistic (e.g., covariance/Gram matrix) $\Sigma \in \mathbb{R}^{d \times d}$, such as computing a factorization of $\Sigma + \lambda I$ or its inverse, and then applying it to obtain a closed-form update. A standard dense factorization (SVD/Cholesky) has $O(d^3)$ time and $O(d^2)$ memory. If a factorization is precomputed and cached, each edit still applies a dense linear map, which costs $O(d^2)$ time, while the stored statistic/factor remains $O(d^2)$ memory. This yields the table entry: $T = O(d^3)$ (or $O(d^2)$ apply) and $M = O(d^2)$.

#### E.4.2. NULL-SPACE CONSTRAINT (SVD ON ACCUMULATED KEYS $K_t$)

Null-space constrained editors maintain constraints w.r.t. accumulated keys. Let $K_t = [k_1, \ldots, k_t] \in \mathbb{R}^{d \times t}$ be the matrix of past keys. Computing an orthogonal complement of $\text{span}(K_t)$ via SVD on the tall matrix $K_t$ typically costs $O(d\,t^2)$ time and requires storing $K_t$ or its bases, i.e., $O(d\,t)$ memory, for $d \gg t$. Hence both time and memory grow with $t$.

#### E.4.3. PARAMETER-PRESERVING MEMORY/EXPERT METHODS

These methods store edit-specific items (e.g., adapters/experts or memory entries). If each edit stores an item of size $p_{\text{mem}}$, then memory grows linearly as $M = O(t\,p_{\text{mem}})$. At inference, selecting among $t$ items costs $\text{Sel}(t)$, commonly $O(t)$ for brute-force scan or $O(\log t)$ (or sublinear) with approximate nearest-neighbor indexing. This justifies the table entry $T = O(\text{Sel}(t))$ and $M = O(t\,p_{\text{mem}})$.

#### E.4.4. NAIVE FINETUNING (LORA RANK $r$)

Under the same low-rank interface (updating $B^{(l)} \in \mathbb{R}^{d_{\text{out}} \times r}$ with SGD/Adam) but without the preconditioner recursion, the editor-specific update is a single step $B^{(l)} \leftarrow B^{(l)} - \eta G_t^{(l)}$, which costs $O(d_{\text{out}}r)$ time per layer. Storing the trainable parameters costs $O(d_{\text{out}}r)$ per layer. Thus, this is $T = O(|\mathcal{L}|\,d_{\text{out}}r)$ and $M = O(|\mathcal{L}|\,d_{\text{out}}r)$, both independent of $t$.

*Table 11.* Complete editing results on E-VQA and E-IC for BLIP2-OPT and LLaVA-v1.5 under different edit horizons. "Rel.", "T/M-Gen.", and "T/M-Loc." abbreviate *Reliability*, *Generality*, and *Locality* (for text/modal evaluations), respectively. The subscript of each method (e.g., $_1$, $_{10}$, $_{100}$) denotes the number of online edits performed. Rows shaded in light purple indicate *parameter-modifying* methods.

| Model | Methods | E-VQA | | | | | | E-IC | | | | | |
|---|---|---|---|---|---|---|---|---|---|---|---|---|---|
| | | Rel. | T-Gen. | M-Gen. | T-Loc. | M-Loc. | Avg. | Rel. | T-Gen. | M-Gen. | T-Loc. | M-Loc. | Avg. |
| **BLIP2-OPT** | FT-L$_1$ | 100.00 | 100.00 | 60.00 | 94.74 | 100.00 | 90.95 | 96.77 | 95.02 | 90.72 | 90.05 | 68.27 | 88.16 |
| | FT-M$_1$ | 100.00 | 96.67 | 63.33 | 100.00 | 73.33 | 86.67 | 100.00 | 100.00 | 76.92 | 100.00 | 24.79 | 80.34 |
| | MEND$_1$ | 100.00 | 100.00 | 100.00 | 60.61 | 33.33 | 78.79 | 100.00 | 100.00 | 100.00 | 84.21 | 100.00 | 96.84 |
| | AlphaEdit$_1$ | 60.00 | 50.00 | 40.00 | 91.64 | 73.33 | 62.99 | 30.77 | 30.77 | 30.77 | 89.47 | 100.00 | 56.35 |
| | SERAC$_1$ | 100.00 | 100.00 | 100.00 | 89.47 | 80.00 | 93.89 | 97.38 | 95.52 | 86.11 | 100.00 | 63.82 | 88.57 |
| | IKE$_1$ | 100.00 | 100.00 | 100.00 | 52.63 | 13.33 | 73.19 | 100.00 | 100.00 | **100.00** | 63.16 | 10.00 | 74.63 |
| | LiveEdit$_1$ | 92.68 | 92.33 | 89.25 | **100.00** | 95.57 | 93.97 | 80.67 | 80.67 | 77.79 | 100.00 | 98.09 | 87.44 |
| | **M-ORE$_1$ (Ours)** | **100.00** | **100.00** | **100.00** | 94.74 | **100.00** | **98.95** | **100.00** | **100.00** | 93.75 | **100.00** | **100.00** | **98.75** |
| | FT-L$_{10}$ | 80.00 | 60.00 | 40.00 | 90.06 | 87.92 | 71.59 | 92.09 | 90.31 | 70.43 | 91.85 | 55.52 | 80.04 |
| | FT-M$_{10}$ | 76.67 | 73.33 | 43.33 | **100.00** | 43.33 | 67.33 | 80.48 | 80.44 | 65.91 | **100.00** | 11.56 | 67.68 |
| | MEND$_{10}$ | 2.33 | 2.33 | 2.67 | 69.41 | 60.00 | 27.35 | 0.00 | 0.00 | 0.00 | 28.48 | 30.00 | 11.70 |
| | AlphaEdit$_{10}$ | 40.83 | 38.33 | 32.50 | 88.19 | 64.17 | 52.80 | 37.35 | 36.17 | 34.30 | 95.39 | 84.81 | 57.60 |
| | SERAC$_{10}$ | 90.40 | 92.05 | 90.62 | 89.00 | 40.06 | 80.43 | 86.65 | 87.49 | 87.22 | 91.24 | 60.48 | 82.62 |
| | IKE$_{10}$ | 91.80 | 91.74 | 90.95 | 73.06 | 8.67 | 71.24 | 80.40 | 80.46 | **90.23** | 73.44 | 11.83 | 67.27 |
| | LiveEdit$_{10}$ | 92.39 | 91.73 | 85.46 | 99.67 | 95.33 | 92.92 | 79.80 | 77.21 | 67.93 | 98.96 | 96.37 | 84.05 |
| | **M-ORE$_{10}$ (Ours)** | **100.00** | **96.67** | **100.00** | 92.73 | **96.67** | **97.21** | **93.58** | **93.44** | 73.16 | 97.78 | **96.67** | **90.93** |
| | FT-L$_{100}$ | 26.00 | 18.00 | 11.00 | 86.28 | 53.12 | 38.88 | 79.45 | 71.69 | 57.82 | 92.23 | 55.18 | 71.27 |
| | FT-M$_{100}$ | 58.40 | 50.85 | 43.69 | **100.00** | 46.85 | 59.96 | 67.18 | 62.17 | 51.63 | **100.00** | 9.81 | 58.16 |
| | MEND$_{100}$ | 1.00 | 1.00 | 1.00 | 90.42 | 77.20 | 34.12 | 0.00 | 0.00 | 0.00 | 46.96 | 54.05 | 20.20 |
| | AlphaEdit$_{100}$ | 36.83 | 28.50 | 26.78 | 86.35 | 69.97 | 49.69 | 27.94 | 28.57 | 26.69 | 90.52 | 78.83 | 50.51 |
| | SERAC$_{100}$ | 88.03 | 85.18 | 88.03 | 89.95 | 37.60 | 77.76 | 74.02 | 63.79 | 56.93 | 77.52 | 42.94 | 63.04 |
| | IKE$_{100}$ | 83.53 | 83.00 | 84.72 | 74.63 | 6.37 | 66.45 | 81.18 | 71.13 | **84.30** | 79.90 | 11.93 | 65.69 |
| | LiveEdit$_{100}$ | 91.83 | 91.16 | 85.02 | 99.31 | **92.78** | 92.02 | 74.63 | 74.33 | 61.95 | 96.77 | 95.34 | 80.60 |
| | **M-ORE$_{100}$ (Ours)** | **96.05** | **92.05** | **94.06** | 91.77 | 88.85 | **92.56** | **83.14** | **78.17** | 60.08 | 95.51 | **95.58** | **82.49** |
| **LLaVA-v1.5** | FT-L$_1$ | 95.66 | 99.00 | 81.84 | 89.38 | 89.98 | 91.17 | 100.00 | 100.00 | 89.80 | 91.67 | 28.01 | 81.89 |
| | FT-M$_1$ | 95.00 | 95.00 | 79.67 | 100.00 | 85.83 | 91.10 | 100.00 | 100.00 | 76.47 | 100.00 | 26.11 | 80.52 |
| | MEND$_1$ | 95.53 | 95.53 | 83.79 | 74.82 | 59.65 | 81.86 | 96.81 | 97.65 | **96.44** | 94.74 | 100.00 | 97.13 |
| | AlphaEdit$_1$ | 72.57 | 72.57 | 70.67 | 88.04 | 96.67 | 80.10 | 47.06 | 47.06 | 52.94 | 100.00 | 100.00 | 69.41 |
| | SERAC$_1$ | 90.00 | 90.00 | 60.00 | 100.00 | 23.33 | 72.67 | 88.61 | 90.17 | 81.45 | 98.24 | 50.83 | 81.86 |
| | IKE$_1$ | 50.00 | 50.00 | 50.00 | 58.33 | 15.00 | 44.67 | 94.12 | 94.12 | 94.12 | 62.50 | 12.50 | 71.47 |
| | LiveEdit$_1$ | 93.36 | 93.67 | **87.91** | 100.00 | 100.00 | 94.99 | 82.33 | 82.33 | 80.67 | 100.00 | 100.00 | 89.07 |
| | **M-ORE$_1$ (Ours)** | **100.00** | **100.00** | 85.12 | **100.00** | **100.00** | **97.02** | **100.00** | **100.00** | 94.12 | **100.00** | **100.00** | **98.82** |
| | FT-L$_{10}$ | 90.00 | 90.00 | 85.00 | 92.93 | 81.82 | 87.95 | 92.41 | 90.93 | 81.01 | 92.04 | 20.02 | 75.28 |
| | FT-M$_{10}$ | 80.00 | 80.00 | 69.67 | **100.00** | 62.41 | 78.42 | 94.50 | 92.96 | 70.12 | 100.00 | 21.41 | 75.80 |
| | MEND$_{10}$ | 9.74 | 9.74 | 10.66 | 78.71 | 60.14 | 33.80 | 1.80 | 1.76 | 0.98 | 5.89 | 8.33 | 3.75 |
| | AlphaEdit$_{10}$ | 74.67 | 73.48 | 71.80 | 88.37 | 91.45 | 79.95 | 48.67 | 48.67 | 52.33 | 93.58 | 85.09 | 65.67 |
| | SERAC$_{10}$ | 87.33 | 86.83 | 68.68 | 97.47 | 26.43 | 73.35 | 80.09 | 77.91 | 62.94 | 75.53 | 22.12 | 63.72 |
| | IKE$_{10}$ | 38.00 | 32.67 | 36.00 | 53.81 | 11.86 | 34.47 | 85.48 | 84.18 | 85.48 | 60.70 | 10.61 | 65.29 |
| | LiveEdit$_{10}$ | 92.75 | 93.05 | 84.17 | 98.67 | **97.93** | 93.31 | 80.77 | 80.80 | 73.98 | 100.00 | **98.71** | 86.85 |
| | **M-ORE$_{10}$ (Ours)** | **100.00** | **100.00** | **88.63** | 99.11 | 95.00 | **96.55** | **98.02** | **97.25** | 85.51 | **100.00** | 90.24 | **94.20** |
| | FT-L$_{100}$ | 74.55 | 66.59 | 66.14 | 84.15 | 65.71 | 71.43 | 86.00 | 82.11 | 78.15 | 77.13 | 11.33 | 66.94 |
| | FT-M$_{100}$ | 85.67 | 81.75 | 67.03 | **100.00** | 46.19 | 76.13 | 84.57 | 80.05 | 62.88 | **100.00** | 8.32 | 67.16 |
| | MEND$_{100}$ | 1.09 | 1.09 | 1.01 | 75.33 | 61.67 | 28.04 | 0.11 | 0.08 | 0.04 | 25.52 | 31.33 | 11.42 |
| | AlphaEdit$_{100}$ | 71.33 | 70.93 | 70.67 | 88.45 | 77.67 | 75.81 | 51.98 | 48.21 | 47.23 | 92.59 | 56.90 | 59.38 |
| | SERAC$_{100}$ | 87.71 | 85.03 | 67.13 | 92.27 | 20.83 | 70.59 | 73.38 | 71.00 | 59.72 | 72.88 | 25.85 | 60.57 |
| | IKE$_{100}$ | 38.87 | 35.85 | 39.40 | 46.22 | 11.24 | 34.32 | 78.78 | 77.37 | 78.07 | 53.86 | 12.88 | 60.19 |
| | LiveEdit$_{100}$ | 90.22 | 91.39 | 81.49 | 98.05 | **95.27** | 91.28 | 78.49 | 78.77 | 65.50 | 98.77 | **96.76** | 83.66 |
| | **M-ORE$_{100}$ (Ours)** | **97.43** | **94.30** | 85.40 | 96.32 | 91.90 | **93.07** | **94.64** | **93.27** | **78.66** | 97.39 | 89.71 | **90.73** |

**Target: Long tables with black chairs behind them.**
**Prompt:** A photo of…
**Answer (Before Editing):** Black tables behind.
**Answer (After Editing) :** Long tables with black chairs behind them.
**Rephrase Prompt:** Describe this picture.
**Answer (Post rephrase) :** Long tables with black chairs behind them.

| Orig. | Base | L1 | L16 | L31 |

**Target: Shopping.**
**Prompt:** Why might someone go to this place?
**Answer (Before Editing):** Someone might go to this place to experience the bustling city life, as it is a crowded street with many people and various activities. The crowded street is filled with people walking, talking on cell phones, and engaging in different activities.
**Answer (After Editing):** Shopping.

| Orig. | Base | L1 | L16 | L31 |

**Target: A very old looking pot is on display.**
**Prompt:** A photo of…
**Answer (Before Editing):** The pot is very old looking and is on display. The pot is very pretty.
**Answer (After Editing):** A very old looking pot is on display.
**Rephrase Prompt:** Describe this picture.
**Answer (Post rephrase):** A very old looking pot is on display.

| Orig. | Base | L1 | L16 | L31 |

**Target: Office.**
**Prompt:** What is this desk used for?
**Answer (Before Editing):** Office.
**Answer (After Editing):** Office.

| Orig. | Base | L1 | L16 | L31 |

*Figure 13.* E-IC case studies (challenging visual-understanding edits). Top two panels: Group 1; bottom two panels: Group 2. In each group, the upper panel is the edit sample and the lower panel is the corresponding locality sample.

**Target: A plane is coming in for a landing.**
**Prompt:** A photo of…
**Answer (Before Editing):** A plane is on the runway.
**Answer (After Editing):** A plane is coming in for a landing.
**Rephrase Prompt:** Describe the content of the image.
**Answer (Post rephrase):** The image is a black and white photo of an airplane's landing gear.

| Orig. | Base | L1 | L16 | L31 |
|---|---|---|---|---|

**Target: Suit.**
**Prompt:** What is the person in the photo wearing?
**Answer (Before Editing):** Suit.
**Answer (After Editing):** Suit.

| Orig. | Base | L1 | L16 | L31 |
|---|---|---|---|---|

**Target: A single train car is passing under a walk bridge.**
**Prompt:** A photo of…
**Answer (Before Editing):** The train car is a single train car passing under a walk bridge.
And the car is a single train car.
**Answer (After Editing):** A single train car is passing under a walk bridge.
**Rephrase Prompt:** Provide a detailed description of the image.
**Answer (Post rephrase):** The image features a single train car passing under a walk bridge.

| Orig. | Base | L1 | L16 | L31 |
|---|---|---|---|---|

**Target: Tipping.**
**Prompt:** What do teenage pranksters do to this animal?
**Answer (Before Editing):** Teenage pranksters often play pranks on cows by tipping over their feeding troughs or spraying them with water. In the image, a group of cows is grazing in a field, and there are a couple of cows that are tipped over.
**Answer (After Editing):** Tipping over a cow.

| Orig. | Base | L1 | L16 | L31 |
|---|---|---|---|---|

*Figure 14.* E-IC case studies (challenging visual-understanding edits). Top two panels: Group 1; bottom two panels: Group 2. In each group, the upper panel is the edit sample and the lower panel is the corresponding locality sample.

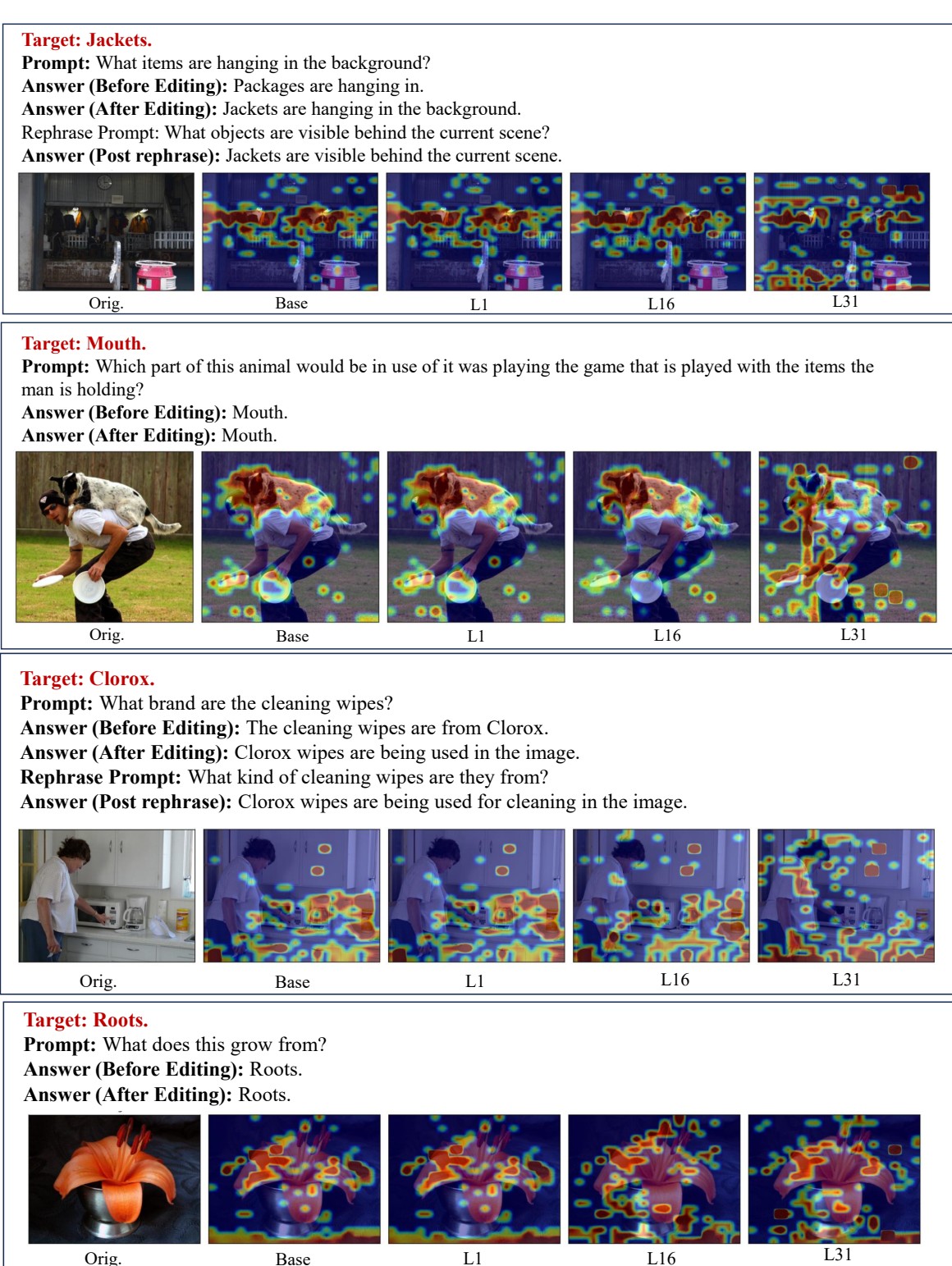

**Target: Jackets.**
**Prompt:** What items are hanging in the background?
**Answer (Before Editing):** Packages are hanging in.
**Answer (After Editing):** Jackets are hanging in the background.
Rephrase Prompt: What objects are visible behind the current scene?
**Answer (Post rephrase):** Jackets are visible behind the current scene.

Orig.   Base   L1   L16   L31

**Target: Mouth.**
**Prompt:** Which part of this animal would be in use of it was playing the game that is played with the items the man is holding?
**Answer (Before Editing):** Mouth.
**Answer (After Editing):** Mouth.

Orig.   Base   L1   L16   L31

**Target: Clorox.**
**Prompt:** What brand are the cleaning wipes?
**Answer (Before Editing):** The cleaning wipes are from Clorox.
**Answer (After Editing):** Clorox wipes are being used in the image.
**Rephrase Prompt:** What kind of cleaning wipes are they from?
**Answer (Post rephrase):** Clorox wipes are being used for cleaning in the image.

Orig.   Base   L1   L16   L31

**Target: Roots.**
**Prompt:** What does this grow from?
**Answer (Before Editing):** Roots.
**Answer (After Editing):** Roots.

Orig.   Base   L1   L16   L31

*Figure 15.* E-VQA case studies. Top two panels: Group 1; bottom two panels: Group 2. In each group, the upper panel is the edit sample and the lower panel is the corresponding locality sample.

**Target: Goodfellas.**
**Prompt:** What Hollywood movie is one of the food dishes named after?
**Answer (Before Editing):** One of the food dishes named after a Hollywood movie is the Pizza Hut pizza. Pizza Hut is a pizza restaurant chain that was inspired by the 1990 movie Pizza Hut starring John Travolta.
**Answer (After Editing):** Goodfellas.
**Rephrase Prompt:** What food dish is named after a Hollywood movie?
**Answer (Post rephrase):** Goodfellas pizza is named after a Hollywood movie.

| Orig. | Base | L1 | L16 | L31 |
|-------|------|-----|------|------|

**Target: Trying to hit the ball.**
**Prompt:** What is that man doing with the bat?
**Answer (Before Editing):** Batting.
**Answer (After Editing):** Trying to hit a ball.

| Orig. | Base | L1 | L16 | L31 |
|-------|------|-----|------|------|

**Target: Mat.**
**Prompt:** What's to the left of the plate?
**Answer (Before Editing):** Pizza.
**Answer (After Editing):** Mat.
**Rephrase Prompt:** What is located to the left of the plate?
**Answer (Post rephrase):** Mat.

| Orig. | Base | L1 | L16 | L31 |
|-------|------|-----|------|------|

**Target: Passenger.**
**Prompt:** What type of plane is that?
**Answer (Before Editing):** Passenger.
**Answer (After Editing):** Passenger plane.

| Orig. | Base | L1 | L16 | L31 |
|-------|------|-----|------|------|

*Figure 16.* E-VQA case studies. Top two panels: Group 1; bottom two panels: Group 2. In each group, the upper panel is the edit sample and the lower panel is the corresponding locality sample.

