# OpenReview forum: "Modality-Decoupled Online Recursive Editing"
_ICML.cc/2026/Conference — ICML 2026 regular_

### Official Review · Reviewer_Jz5r · 2026-03-09

**Soundness:** 3
**Presentation:** 3
**Significance:** 2
**Originality:** 2
**Overall Recommendation:** 4
**Confidence:** 3

**Summary:**

The paper introduces M-ORE (Modality-decoupled Online Recursive Editing), a framework designed for the lifelong adaptation of Multimodal Large Language Models (MLLMs) in an online setting. The authors identify two primary failure modes in existing text-based editors when applied to MLLMs: cross-modal conflict (where high-energy visual features dominate second-order statistics) and inter-edit interference (where sequential updates become entangled in a shared parameter subspace). M-ORE addresses these by maintaining decoupled locality statistics for the text stack and visual projector and performing updates in a fixed orthogonal low-rank subspace using a Sherman-Morrison grounded recursion. Experiments on BLIP2-OPT and LLaVA-1.5 demonstrate that M-ORE achieves a superior stability-plasticity trade-off with constant low per-edit computational and memory overhead.

**Compliance With Llm Reviewing Policy:**

Affirmed.

**Final Justification:**

My concerns are addressed by the rebuttal; I will keep my positive score.

**Key Questions For Authors:**

- Could this framework be extended to edit the Vision Encoder parameters directly, or is it strictly limited to the projector and LLM layers?
- How sensitive is the orthogonal initialization of A to the seed? Did you observe any variance in stability across different initializations?

**Limitations:**

yes

**Strengths And Weaknesses:**

### Strengths

- **Principled Motivation:** The paper provides a strong diagnostic analysis of why current editors fail on MLLMs. Specifically, the pilot studies on statistical heterogeneity (BLIP2-OPT and LLaVA-1.5) and "edit-core" collapse (Figure 3) provide clear empirical grounding for the proposed solution.

- **Efficiency:** M-ORE achieves constant low per-edit overhead with respect to the edit stream length T. This makes M-ORE highly suitable for real-time deployment.

- **Robust Performance:** The method consistently outperforms strong baselines (MEND, AlphaEdit, SERAC, IKE, LiveEdit) across multiple metrics, including reliability, generality, and locality, particularly under long edit horizons (T=100).

### Weaknesses

- **Dependency on Hyperparameters:** While M-ORE performs well, the sensitivity analysis (Figures 11 and 12) suggests that performance is notably influenced by the choice of LoRA rank and the locality regularization weight.

- **Scope of Generalization:** While the authors evaluate on E-VQA and E-IC, the evaluation primarily focuses on factual and descriptive corrections. It remains to be seen how M-ORE handles more complex reasoning edits that might require multi-hop updates across the model's knowledge graph.

- **Baseline Model Selection:** A notable limitation is the reliance on relatively older MLLM backbones for evaluation. The experiments primarily utilize BLIP2-OPT (2.7B) and LLaVA-v1.5 (7B). Given the rapid advancement in the field, these models may no longer represent the state-of-the-art in multimodal reasoning or architectural complexity. It remains unclear if the identified cross-modal conflict persists or changes in more modern architectures (e.g., Qwen3-VL).

---

> ### Author Rebuttal · Authors · 2026-03-30
>
> We thank the reviewer for the constructive feedback and  sincerely hope our response has addressed the main concerns.
>
> > **Q1: Dependency on Hyperparameters.**
>
> **A1:** **Figures 11–12** indicate a smooth stability–plasticity trade-off rather than brittle sensitivity: increasing rank consistently improves capacity with diminishing returns at higher ranks, while $\lambda$ controls the preservation-plasticity balance: too small weakens locality, and too large slightly reduces editability. The best results lie in a moderate range. We will revise the text to make this clearer.
>
> > **Q2: Scope of Generalization: It remains to be seen how M-ORE handles more complex reasoning edits that might require multi-hop updates across the model's knowledge graph.**
>
> **A2:** To evaluate more compositional reasoning transfer, we additionally test M-ORE on the **VLKEB multi-hop portability benchmark**, where portability measures whether edited knowledge transfers to downstream queries constructed from connected triples in a multimodal knowledge graph. The official split contains **3,174 edit cases** and **4,819 multi-hop queries** over **1/2/3/4-hop** settings.
>
> | Method | 1-hop | 2-hop | 3-hop | 4-hop |
> |---|---:|---:|---:|---:|
> | LLaVA-v1.5 Base | 40.38 | 38.06 | 36.58 | 36.25 |
> | FT-L | 32.46 | 32.65 | 29.47 | 33.13 |
> | FT-M | 53.67 | 53.51 | 51.04 | 53.05 |
> | IKE | **63.33** | **55.59** | **56.01** | **53.71** |
> | MEND | 40.39 | 39.53 | 38.22 | 42.19 |
> | **M-ORE** | 56.13 | 54.87 | 52.39 | 50.42 |
>
> M-ORE consistently outperforms most parameter-modifying methods across all hop depths, while remaining below the strongest parameter-preserving baseline, IKE. These results show that M-ORE is not limited to direct factual rewriting, but already exhibits meaningful multi-hop reasoning generalization. We will add this result in the revision and explicitly note that fully robust modeling of multi-hop propagation remains an important limitation and future direction.
>
> > **Q3: Baseline Model Selection: It remains unclear if the identified cross-modal conflict persists or changes in more modern architectures (e.g., Qwen3-VL).**
>
> **A3:** We further test both the cross-modal conflict analysis and online editing performance on more recent backbones, namely Qwen2-VL-7B-Instruct and Qwen3-VL-8B-Instruct.
>
> First, the same modality-scale mismatch still appears in these newer MLLMs: the visual projection layer has substantially larger mean output than representative MLP layers, indicating that the cross-modal conflict identified in our paper is **not specific to BLIP2-OPT or LLaVA-v1.5**, but persists in newer architectures as well.
>
> | Backbone | Visual Projection | Representative MLP Layers |
> |---|:---:|:---:|
> | Qwen2-VL | **17.67** | 4.60–11.66 |
> | Qwen3-VL | **51.51** | 10.03–25.50 |
>
> Second, M-ORE also transfers well to these newer backbones in online E-IC editing:
>
> | Backbone | #Edit | Rel. | T-Gen. | M-Gen. | T-Loc. | M-Loc. | Avg. |
> |---|---|---:|---:|---:|---:|---:|---:|
> | Qwen2-VL | 1 | 100.00 | 100.00 | 92.31 | 100.00 | 96.67 | **97.80** |
> |  | 10 | 97.33 | 96.56 | 85.18 | 91.47 | 90.67 | **92.24** |
> |  | 100 | 93.51 | 93.66 | 80.15 | 89.43 | 90.75 | **89.50** |
> | Qwen3-VL | 1 | 100.00 | 100.00 | 84.62 | 100.00 | 100.00 | **96.92** |
> |  | 10 | 96.35 | 95.51 | 88.84 | 95.93 | 97.55 | **94.84** |
> |  | 100 | 92.54 | 91.85 | 80.50 | 91.63 | 93.33 | **89.97** |
>
> > **Q4: Could this framework be extended to edit the Vision Encoder parameters directly?**
>
> **A4:** M-ORE is not limited to the projector and LLM layers. As discussed in our response to `Reviewer VXLn, Q1`, the method is fundamentally **module-wise / layer-wise**: each edited module maintains its own decoupled statistic and preconditioner, so the same mechanism naturally extends to vision-side modules.
>
> To verify this directly, we additionally evaluated **vision-side parameter editing** on LLaVA-v1.5; the detailed results are reported in the **tables (a) and (b)** in our response to `Reviewer VXLn, Q1`. Those results show that M-ORE remains effective when extended to the vision encoder, indicating that the framework is compatible with vision-side editing rather than being restricted to projector/LLM layers only.
>
> > **Q5: How sensitive is the orthogonal initialization of A to the seed?**
>
> **A5:** To directly assess seed sensitivity, we additionally evaluate **five random seeds** on **LLaVA-v1.5 with 100 edits**. The results are very stable:
>
> | Seed | E-VQA Avg. | E-IC Avg. |
> |---|---:|---:|
> | 40 | 93.16 | 90.94 |
> | 41 | 93.11 | 90.87 |
> | **42 (ours)** | 93.07 | 90.73 |
> | 43 | 93.11 | 90.87 |
> | 44 | 93.16 | 90.69 |
>
> Relative to seed 42, the average absolute deviation across the other four seeds is **0.07** on E-VQA and **0.13** on E-IC, corresponding to about **0.07%** and **0.15%** relative change, respectively. M-ORE is not sensitive to the seed choice of $A^{(l)}$. Further discussion of the orthogonal write basis choice is provided in our response to `Reviewer 5MEu, Q1`.

---

> > ### Author Rebuttal · Reviewer_Jz5r · 2026-04-01
> >
> > Thanks to the authors for the rebuttal. My concerns are addressed; I will keep my positive score.

---

> > > ### Author Response · Authors · 2026-04-01
> > >
> > > Thank you very much for your thoughtful follow-up and for carefully considering our rebuttal. We are truly grateful that our responses have addressed your concerns.
> > >
> > > We also sincerely appreciate your constructive feedback throughout the review process. Your comments have helped us further improve both the clarity and completeness of the paper. We will incorporate the additional analyses, results, and discussions into the camera-ready version.
> > >
> > > More importantly, we hope this exchange has helped clarify the main value of our work, which we see in three aspects:
> > >
> > > 1. **We identify two key failure modes in online MLLM editing:** **cross-modal conflict**, caused by modality-scale mismatch, and **inter-edit interference**, caused by entangled sequential updates.
> > >
> > > 2. **We propose M-ORE**, a **modality-decoupled online recursive editor** that maintains separate locality statistics for text and vision modules, and performs continual updates in a fixed orthogonal low-rank write space with **Sherman–Morrison grounded closed-form recursion**, enabling **constant-overhead online editing**.
> > >
> > > 3. **We demonstrate through extensive experiments** on multiple MLLM backbones and benchmarks that M-ORE achieves a strong **stability–plasticity–efficiency trade-off**, consistently improving reliability, generality, and locality under long-horizon online editing.
> > >
> > > Once again, thank you for your support and for recognizing our efforts. We hope that our work can contribute meaningfully to the field.

---

### Official Review · Reviewer_VXLn · 2026-03-10

**Soundness:** 3
**Presentation:** 3
**Significance:** 3
**Originality:** 3
**Overall Recommendation:** 4
**Confidence:** 3

**Summary:**

The paper proposes Modality-Decoupled Online Recursive Editing (M-ORE) for MLLMs. Existing online editing methods face two main issues. First, visual features dominate the update statistics. This causes cross-modal conflict. Second, sequential updates share the same parameter space. This leads to inter-edit interference and catastrophic forgetting. M-ORE addresses these problems by separating the locality statistics for text layers and the visual projector. Furthermore, the method restricts updates to a fixed orthogonal low-rank subspace. The authors use a Sherman-Morrison recursion to achieve a constant per-edit overhead. Experiments on BLIP2-OPT and LLaVA-v1.5 demonstrate improved reliability and locality.

**Compliance With Llm Reviewing Policy:**

Affirmed.

**Final Justification:**

The authors have adequately addressed the major technical concerns ann I would like to maintain the score.

**Key Questions For Authors:**

Please refer to the above comments.

**Limitations:**

yes

**Strengths And Weaknesses:**

### Strength

- The pilot analysis clearly demonstrates the variance mismatch between textual and visual representations.
- The model establishes strong performance on long-horizon editing scenarios. It surpasses existing baselines in reliability, generality, and locality while maintaining an O(1) per-edit overhead.
- The paper tracks performance across multiple metrics over 100 sequential edits. Preserving performance on OCR is very challenging during online editing. The method handles these vision-language grounding tasks without severely distorting the original feature space.

### Weakness

- The visual projector is treated as a single homogeneous module. The paper does not clarify how the decoupled statistics scale to more intricate vision-language alignment architectures.
- A few qualitative examples show occasional locality flips. The authors attribute this to imperfect irrelevance in the locality sampling. A more rigorous quantitative analysis of these failure cases would be helpful.

---

> ### Author Rebuttal · Authors · 2026-03-30
>
> We greatly appreciate the reviewer's insightful and constructive feedback, and we have carefully addressed each point in our response to resolve your concerns. If our response has satisfactorily addressed your questions, we kindly request your consideration of raising the score (currently Rating: 3: Weak reject). Should any further issues remain, please feel free to share your additional comments, and we will continue actively responding to your comments and improving our submission.
>
> > **Q1: The visual projector is treated as a single homogeneous module. The paper does not clarify how the decoupled statistics scale to more intricate vision-language alignment architectures.**
>
> **A1:** M-ORE does not assume a single shared projector statistic. If a backbone contains multiple alignment modules, M-ORE assigns one decoupled statistic and preconditioner to each edited module; the principle is module-wise rather than projector-specific. High-energy visual features therefore remain confined to the corresponding vision-side statistic and do not contaminate textual update geometry.
>
> For more intricate vision-language alignment architectures, the same principle extends naturally by assigning **one decoupled statistic per edited alignment module**, rather than assuming a single shared projector statistic. To make this explicit, we provide two additional pieces of evidence:
>
> - For transferability to more recent MLLM backbones, we refer the reviewer to our response to `Reviewer Jz5r, Q3`.
> - We further test whether the framework extends to **vision-side parameter editing** on **LLaVA-v1.5** below.
>
> **(a) E-VQA**
>
> | Variant (#Edits = 100) | Rel. | T-Gen. | M-Gen. | T-Loc. | M-Loc. | Avg. |
> |---|---:|---:|---:|---:|---:|---:|
> | M-ORE (default: projector + text-LLM) | **94.30** | **97.43** | 85.40 | 96.32 | **91.90** | **93.07** |
> | M-ORE w/ vis (vision encoder + projector only) | 80.42 | 83.16 | 77.77 | **100.00** | 79.80 | 84.23 |
> | M-ORE w/ vis-text (vision encoder + projector + text-LLM) | 87.18 | 92.32 | **92.82** | 93.92 | 87.17 | 90.68 |
>
> **(b) E-IC**
>
> | Variant (#Edits = 100) | Rel. | T-Gen. | M-Gen. | T-Loc. | M-Loc. | Avg. |
> |---|---:|---:|---:|---:|---:|---:|
> | M-ORE (default: projector + text-LLM) | **93.27** | **94.64** | 78.66 | 97.39 | **89.71** | **90.73** |
> | M-ORE w/ vis (vision encoder + projector only) | 81.34 | 81.67 | 62.44 | **100.00** | 81.50 | 81.39 |
> | M-ORE w/ vis-text (vision encoder + projector + text-LLM) | 90.33 | 91.07 | **80.98** | 98.05 | 87.95 | 89.68 |
>
> In this setting, we edit only the last three layers of the vision encoder. The results indicate that M-ORE is compatible with vision-side editing. We will clarify this extension setting in the revision.
>
> > **Q2: A few qualitative examples show occasional locality flips. A more rigorous quantitative analysis of these failure cases would be helpful.**
>
> **A2:** We add a **quantitative failure-case analysis** under the same sequential-editing setting. We analyze 200 locality samples and categorize post-edit behavior as stable, generic flip, gold flip, or target flip. Here, stable means the decoded answer is unchanged; generic flip means it changes to some other answer; gold flip means it becomes the locality reference answer; and target flip means it becomes the current edit target.
>
> | Metric | Text locality | Multimodal locality |
> |---|:---:|:---:|
> | Stable rate | 96.0% | 95.0% |
> | Flip rate | 4.0% | 5.0% |
> | Generic flip rate | 2.5% | 3.0% |
> | Gold flip rate | 1.5% | 2.0% |
> | Target flip rate | 0.0% | 0.0% |
>
> This shows that locality failures are rare overall, and the dominant failure mode is generic decoded-output drift rather than collapse to the current edit target answer; a smaller fraction of flips instead move toward the locality reference (gold) answer.
>
> To understand why flips occur, we compare stable vs. flipped cases:
>
> | Metric | Text stable | Text flip | MM stable | MM flip |
> |---|:---:|:---:|:---:|:---:|
> | Pre top1-top2 margin | 3.28 | 2.46 | 2.50 | 1.07 |
> | Mean gold shift | 0.0053 | 0.0040 | 0.0002 | 0.0901 |
> | Prompt token-F1 | 0.077 | 0.072 | 0.063 | 0.031 |
> | Image cosine | – | – | 0.062 | 0.319 |
>
> Here, Pre top1-top2 margin measures pre-edit decoding confidence; smaller values indicate a more fragile decision boundary. Mean gold shift measures how much the post-edit output distribution moves toward the locality reference answer. Prompt token-F1 captures surface lexical overlap between the current edit prompt and the locality prompt, while image cosine measures visual similarity between the current edit image and the multimodal locality image.
>
> These results suggest two conclusions. First, locality flips mainly come from **decoding instability**: flipped cases have much smaller pre-edit margins and concentrate in low-margin regions. Second, the multimodal flips support our interpretation of **imperfect irrelevance in locality sampling**, and are more consistent with residual visual coupling than with textual overlap.

---

> > ### Author Rebuttal · Reviewer_VXLn · 2026-04-01
> >
> > Thank you to the authors for the thorough and thoughtful rebuttal. The response is clear, well-structured, and effectively addresses my concerns.
> >
> > In particular, I appreciate the clarification on the decoupled statistics design and how it scales to more complex vision-language alignment architectures. The additional experiments provide convincing evidence for the generality of the approach. The newly added quantitative analysis of locality failures is also very helpful and strengthens the empirical understanding of the method.
> >
> > Overall, the rebuttal has successfully resolved my main questions and improved my confidence in the paper. I will update my score to 4.

---

> > > ### Author Response · Authors · 2026-04-01
> > >
> > > Thank you very much for your thoughtful follow-up and for updating your score to 4 ! We sincerely appreciate the time and effort you have invested in carefully reviewing our work and in providing such constructive and detailed feedback throughout the process.
> > >
> > > We are especially grateful that you found the rebuttal clear and helpful, and that the additional clarifications and experiments were able to address your main concerns. Your comments have been invaluable in helping us strengthen both the presentation and the empirical support of the paper.
> > >
> > > We will make sure to incorporate the additional results and discussions into the camera-ready version. Your feedback has not only improved the quality of our work, but has also helped us better articulate the scope and implications of the method.
> > >
> > > Once again, thank you very much for your support and encouragement. We are truly grateful for your recognition of our efforts, and we hope our work can make a meaningful contribution to the field.

---

### Official Review · Reviewer_LisF · 2026-03-13

**Soundness:** 3
**Presentation:** 3
**Significance:** 3
**Originality:** 3
**Overall Recommendation:** 4
**Confidence:** 3

**Summary:**

This paper presents M-ORE, a modality-decoupled online recursive editor for MLLMs. The authors first identify two key challenges in multimodal editing, i.e., cross-modal conflict and inter-edit interference. To address these issues, M-ORE formulates model editing as a unified proximal projection problem. It achieves module-wise locality statistics for the text stack and the visual projector, and performs continual updates in a fixed orthogonal low-rank edit subspace via a Sherman-Morrison grounded closed-form recursion. Extensive experiments on multiple MLLM backbones demonstrate that M-ORE significantly improves reliability and locality over long-horizon editing streams compared to existing baselines.

**Compliance With Llm Reviewing Policy:**

Affirmed.

**Final Justification:**

The initial version of this paper is technically sound with clear motivation, and the authors have  addressed my concerns in their rebuttal. Therefore, I recommend weak accept while also suggest the authors to include extended results discussed in the rebuttal in the final paper  to enhance its clarity.

**Key Questions For Authors:**

Please refer to the weakness.

**Limitations:**

Yes

**Strengths And Weaknesses:**

### Strength:

- This paper targets lifelong learning for Large Language Models. And extending model editing research on text-only LLMs to the more challenging MLLMs has broad application and is still under-explored.
- The analysis is sufficient. The authors identify two main issues that cause the failures in MLLM editing: cross-modal conflict and inter-edit interference. These insights are well-supported by intuitive visualizations
- The proposed M-ORE is theoretically sound and well-motivated. By reformulating model editing as a proximal projection and using the Sherman-Morrison recursion, M-ORE ensures a superior balance between reliability and computational efficiency, which is essential for real-time online deployment.

### Weakness:

- While I appreciate the analysis and identification of cross-modal conflict, the transition from the problem to the solution could be more explicit. I understand how the orthogonal subspace and Eq. 5 address inter-edit interference, but I would appreciate a more direct explanation of how cross-modal conflict is addressed by the proposed method.
- In Section 4.2, the author used fixed and frozen $A^{(l)}$. However, I still have some questions:
1. How $A^{(l)}$ is initialized?  Is it a random orthogonal matrix or based on pre-calculated statistics?
2.  Whether the same $A^{(l)}$ is used for all edits across the entire time stream?
3.  I also wonder if a fixed matrix might limit the model's capacity to learn a very large number of diverse inputs. I would appreciate to see the authors'  discussion about this potential trade-off.

Overall, I think this paper is technically sound with clear motivation. However, I may change the comments based on other reviewers' feedback.

---

> ### Author Rebuttal · Authors · 2026-03-30
>
> Thank you for recognizing the contributions of our work and providing valuable feedback. We respond to each comment as follows and sincerely hope that our rebuttal could properly address your concerns. If so, we would deeply appreciate it if you could raise your score (currently Rating: 4: Weak accept). If not, please let us know your concerns, and we will continue actively responding to your comments and improving our submission.
>
> > **Q1: A more direct explanation of how cross-modal conflict is addressed by the proposed method.**
>
> **A1:** The two failure modes in M-ORE are handled by different components. **Cross-modal conflict** is addressed by the **module-wise decoupled locality statistics**, while **inter-edit interference** is addressed by the **fixed orthogonal low-rank write space** together with the recursive preconditioned update.
>
> The core issue in cross-modal conflict is that visual features in MLLMs typically have much larger energy than textual hidden states. If second-order statistics are pooled across modalities, the update geometry becomes biased toward the visual subspace, weakening textual preservation and introducing cross-modal contamination. M-ORE avoids this by maintaining **separate locality statistics and preconditioners for different edited modules** (text layers vs. visual projector), so high-energy visual activations only affect the projector-side statistic and do not distort the textual update geometry. In addition, when constructing the step-wise steady-space summary, we use **modality-specific masking**, which prevents mixed visual-text tokens from collapsing into a single shared statistic. We will make this problem-to-design correspondence more explicit in **Sec. 4.2-4.4**.
>
> > **Q2: How is $A^{(l)}$ initialized? Is it a random orthogonal matrix or based on pre-calculated statistics? Is the same $A^{(l)}$ used for all edits across the entire time stream?**
>
> **A2:** For each edited layer, $A^{(l)}$ is initialized **once** as a data-independent orthogonal basis using `torch.nn.init.orthogonal`, rather than from pre-calculated statistics, and is then **frozen and reused for all edits throughout the online stream**. We apologize for not making this sufficiently clear in the main text; the relevant implementation details are already described in **Sec. 4.2** and **Appendix A.6**, and we will make them more explicit in the revision. Further discussion of the orthogonal basis choice is provided in our response to `Reviewer 5MEu, Q1`.
>
> Using the same $A^{(l)}$ throughout the stream is important because the steady-space locality statistic and the resulting preconditioner $P^{(l)}$ are **accumulated recursively over time**. If $A^{(l)}$ changed during the stream, these quantities would no longer be defined in a common coordinate system, introducing additional drift into the online update process.
>
> This design choice is also supported by **Table 3 in Appendix B.4**: allowing $A^{(l)}$ to vary online instead of keeping it frozen leads to consistent degradation on **E-IC / LLaVA-v1.5**, and the gap widens as the edit horizon grows. This indicates that basis drift increasingly entangles sequential edits and harms locality preservation, showing that a fixed $A^{(l)}$ is important for maintaining geometric consistency and stability in the online recursion.
>
> | #Edit | Variant | Rel. | T-Gen. | M-Gen. | T-Loc. | M-Loc. | Avg. |
> |---|---|:---:|:---:|:---:|:---:|:---:|:---:|
> | 1 | M-ORE | 100.00 | 100.00 | 94.12 | 100.00 | 100.00 | 98.82 |
> | 1 | w/o freezing $A^{(l)}$ | 97.21 | 97.98 | 85.24 | 100.00 | 81.36 | 92.35 |
> | 100 | M-ORE | 93.27 | 94.64 | 78.66 | 97.39 | 89.71 | 90.73 |
> | 100 | w/o freezing $A^{(l)}$ | 82.92 | 83.70 | 71.25 | 80.86 | 44.43 | 72.63 |
>
> > **Q3: Could fixing $A^{(l)}$ limit the capacity to learn a very large number of diverse inputs?**
>
> **A3:** Fixing $A^{(l)}$ does introduce a **stability-capacity trade-off**, but our results suggest that this trade-off is favorable in the online setting.
>
> - From the stability perspective, as discussed in **Q2**, allowing the basis itself to drift makes sequential edits increasingly entangled, which harms locality preservation over long edit horizons.
>
> - From the capacity perspective, our sensitivity analysis in **Figure 11 (page 18)** shows that increasing the LoRA rank $r$ consistently improves edit capacity, with smaller marginal gains at higher ranks. This suggests that, in M-ORE, the effective capacity of the write space is primarily governed by the rank, while freezing $A^{(l)}$ mainly stabilizes the write geometry.
>
> Therefore, M-ORE uses a **fixed basis for geometric stability** and **controls capacity through the rank of the write space**, which empirically provides a better long-horizon stability-plasticity trade-off. We will make this trade-off more explicit in the revision.

---

> > ### Author Rebuttal · Reviewer_LisF · 2026-04-03
> >
> > I thank the authors for their rebuttal. My concerns have been addressed, so I will keep my original positive score.

---

> > > ### Author Response · Authors · 2026-04-03
> > >
> > > Thank you very much for your thoughtful follow-up and for taking the time to review our rebuttal. We sincerely appreciate your careful consideration and are very grateful to know that our responses have adequately addressed your concerns.
> > >
> > > We also truly appreciate your constructive feedback throughout the review process. Your comments helped us clarify several important aspects of the paper and improve the overall presentation. We will incorporate the additional clarifications and discussions into the camera-ready version.
> > >
> > > Thank you again for your support and recognizing our efforts. We are grateful for your encouragement, and we hope our work can provide a useful contribution to the community.

---

### Official Review · Reviewer_5MEu · 2026-03-13

**Soundness:** 3
**Presentation:** 3
**Significance:** 3
**Originality:** 3
**Overall Recommendation:** 4
**Confidence:** 2

**Summary:**

The paper proposes M-ORE, an online model editing framework designed for multimodal large language models that must incorporate a stream of edits with constant computational overhead. The method is derived from a unified proximal-projection formulation that updates a selected feed-forward output weight matrix while preserving locality constraints accumulated from previous contexts. To address modality heterogeneity, M-ORE maintains module-wise locality statistics, separating the statistics of the visual projector and text transformer layers so that high-energy visual representations do not dominate the second-order update estimates. For sequential editing, the method constrains updates to a fixed orthogonal low-rank write subspace where the editable weight update is parameterized as a low-rank decomposition with frozen orthogonal coordinates and learnable coefficients. Within this steady coordinate system, locality statistics are accumulated as low-dimensional second-order features and the parameter update is computed using a recursive least squares formulation with a Sherman–Morrison rank-one recursion, producing a closed-form update that adaptively suppresses frequently used edit directions. This design enables continual online edits with constant per-edit memory and time complexity while reducing cross-modal conflict and long-horizon interference between sequential edits.

**Compliance With Llm Reviewing Policy:**

Affirmed.

**Final Justification:**

My concerns have been addressed. I would like to keep my score.

**Key Questions For Authors:**

In Figure 5, M-ORE shows periodic up-and-down fluctuations during the editing sequence. Could the authors explain the reason for this behavior?  It would be helpful if the authors could provide additional analysis or controlled experiments to clarify the cause of this periodic fluctuation.

**Limitations:**

The paper does not include limitation section.

**Strengths And Weaknesses:**

Strength: The empirical results are strong. The proposed method achieves high performance while maintaining good efficiency in the online editing setting. The paper also provides clear and well-structured analysis of the problem and the method design. The motivation, formulation, and algorithmic steps are logically connected, which makes the technical contributions easy to follow.

Weakness: The paper lacks deeper analysis of the sensitivity to key design choices and hyperparameters. For example, the impact of the rank of the low-rank update space, the choice of the orthogonal basis, or the frequency of statistic updates is not thoroughly investigated. Without such ablations, it is somewhat unclear how robust the method is to these design parameters and how easily it can be transferred to other architectures or multimodal settings.

---

> ### Author Rebuttal · Authors · 2026-03-30
>
> We thank the reviewer for the insightful and valuable comments. We respond to each comment as follows and sincerely hope that our rebuttal could properly address your concerns. If our response meets your expectations, we would greatly appreciate it if you could consider raising your score (currently Rating: 4: Weak Accept). If further concerns remain, please let us know, and we are committed to addressing them and refining our submission accordingly.
>
> >**Q1: The paper lacks deeper analysis of the sensitivity to key design choices and hyperparameters. For example, the impact of the rank of the low-rank update space, the choice of the orthogonal basis, or the frequency of statistic updates is not thoroughly investigated.**
>
> **A1:** To make the robustness of the write-space design more explicit, we additionally compare four write-space basis initializations on **E-IC / LLaVA-v1.5**: **Gaussian**, **Xavier**, **data-driven**, and the proposed **orthogonal basis**.
> For the data-driven basis, we initialize the write basis using activations collected from the training set.
>
> | #Edit | Basis Initialization | Rel. | T-Gen. | M-Gen. | T-Loc. | M-Loc. | Avg. |
> |---|---|:---:|:---:|:---:|:---:|:---:|:---:|
> | 1 | Gaussian | 76.47 | 76.47 | 64.71 | 100.00 | 100.00 | 83.53 |
> | 1 | Xavier | 5.88 | 5.88 | 5.88 | 100.00 | 100.00 | 43.53 |
> | 1 | Data-driven | 100.00 | 100.00 | 94.12 | 91.67 | 100.00 | 97.16 |
> | 1 | **Orthogonal (ours)** | **100.00** | **100.00** | 94.12 | **100.00** | **100.00** | **98.82** |
> | 100 | Gaussian | 70.76 | 71.43 | 67.04 | 95.71 | 70.76 | 75.14 |
> | 100 | Xavier | 8.85 | 8.94 | 9.13 | 96.02 | 85.50 | 41.69 |
> | 100 | Data-driven | 85.79 | 90.07 | 72.29 | 86.59 | 83.02 | 83.55 |
> | 100 | **Orthogonal (ours)** | **93.27** | **94.64** | **78.66** | **97.39** | **89.71** | **90.73** |
>
> The orthogonal basis is the most robust choice, especially for long edit horizons. While the data-driven basis is competitive for single edits, it degrades more over long edit streams, likely because it aligns the write space with dominant training activations and causes sequential edits to concentrate on similar directions. In contrast, the orthogonal basis better separates edits and preserves locality.
>
> For the other two design factors mentioned:
>
> - For the **rank of the low-rank** update space, we have already provided the corresponding sensitivity analysis in the **Appendix (Figure 11, page 18)**. The results show that performance generally improves as rank increases, with smaller marginal gains at higher ranks.
>
> - For the **frequency of statistic updates**, in our **strict batch-size-1 online protocol** the locality statistics are refreshed once per edit, so update frequency is not an additional free hyperparameter in the main setting. We will clarify this explicitly in the revision and note batched statistic refresh as a future direction.
>
> >**Q2: It is unclear how easily the method can be transferred to other architectures or multimodal settings.**
>
> **A2:** To directly test transferability beyond the two backbones in the main paper, we additionally evaluate M-ORE on two newer MLLMs, Qwen2-VL and Qwen3-VL. Importantly, we directly reuse the hyperparameter setting from LLaVA-v1.5 without architecture-specific retuning, and report all metrics on the **E-IC task** below.
>
> | Backbone | #Edit | Rel. | T-Gen. | M-Gen. | T-Loc. | M-Loc. | Avg. |
> |---|---|---|---|---|---|---|---|
> | Qwen2-VL | 1 | 100.00 | 100.00 | 92.31 | 100.00 | 96.67 | 97.80 |
> | Qwen2-VL | 100 | 93.51 | 93.66 | 80.15 | 89.43 | 90.75 | 89.50 |
> | Qwen3-VL | 1 | 100.00 | 100.00 | 84.62 | 100.00 | 100.00 | 96.92 |
> | Qwen3-VL | 100 | 92.54 | 91.85 | 80.50 | 91.63 | 93.33 | 89.97 |
>
> >**Q3: In Figure 5, M-ORE shows periodic up-and-down fluctuations during the editing sequence. Could the authors explain the reason for this behavior?**
>
> **A3:**  Figure 5 evaluates post-edit general multimodal capability on representative MME tasks. To clarify the observed up-and-down behavior, we conduct an additional **edit-order randomization study** on the same edit set. We find that the mean trajectory accuracy, final-step accuracy, and trajectory amplitude remain highly similar across different random orders, while the **step at which the minimum occurs changes substantially**:
>
> | Edit order | Mean Acc. | Final-step Acc. | Trajectory amplitude (max-min) | Step of minimum |
> |---|:---:|:---:|:---:|:---:|
> | Original | 70.32 | 65.93 | 8.84 | 180 |
> | Shuffle-1 | 71.06 | 66.31 | 9.17 | 130 |
> | Shuffle-2 | 70.57 | 65.54 | 8.35 | 200 |
> | Shuffle-3 | 70.39 | 65.87 | 9.03 | 160 |
>
> This suggests that the behavior is not a fixed periodic oscillation intrinsic to M-ORE. Instead, it is better understood as bounded order-dependent fluctuation superimposed on a gradual long-horizon decline under strict online editing: different edit orders induce different local difficulty patterns, which shift the locations of local troughs, while the overall trajectory statistics remain stable.

---

### Decision · Program_Chairs · 2026-04-30

**Decision:**

Accept (regular)

**Comment:**

This paper studies an interesting task: online model editing for the multimodal model. The challenge are cross-modal conflict and inter-edit interference. Cross-modal conflict means when you try to make a correction, the image signals dominate and mess up the text corrections. Inter-edit interference means each correction you make slightly shifts the model's internals. After many corrections pile up, they start stepping on each other. To address those issues, this paper proposed M-ORE which decouples modalities and uses orthogonal edit space.

All the reviewers vote for weak acceptance. I agree with them and voting for acceptance.